# Equivariant spatio-hemispherical networks for diffusion MRI deconvolution

**Axel Elaldi**
New York University
axel.elaldi@nyu.edu

**Guido Gerig**
New York University
gerig@nyu.edu

**Neel Dey**
MIT CSAIL
dey@csail.mit.edu

## Abstract

Each voxel in a diffusion MRI (dMRI) image contains a spherical signal corresponding to the direction and strength of water diffusion in the brain. This paper advances the analysis of such spatio-spherical data by developing convolutional network layers that are equivariant to the $\mathbf{E(3)} \times \mathbf{SO(3)}$ group and account for the physical symmetries of dMRI including rotations, translations, and reflections of space alongside voxel-wise rotations. Further, neuronal fibers are typically antipodally symmetric, a fact we leverage to construct highly efficient spatio-*hemispherical* graph convolutions to accelerate the analysis of high-dimensional dMRI data. In the context of sparse spherical fiber deconvolution to recover white matter microstructure, our proposed equivariant network layers yield substantial performance and efficiency gains, leading to better and more practical resolution of crossing neuronal fibers and fiber tractography. These gains are experimentally consistent across both simulation and in vivo human datasets.

## 1 Introduction

Instead of scalar intensities, each voxel of a diffusion MR image (dMRI) contains spatio-angular measurements of local water diffusion [64]. As in Fig. 1, this yields *spatio-spherical* images living on $\mathbb{R}^3 \times \mathcal{S}^2$ that are used to map neuronal organization *in vivo* [7, 44] alongside several other biomedical use cases [35, 37, 56, 65, 71]. Despite its potential, deriving neuronal fiber pathways using dMRI is hampered by significant partial voluming in both spatial and spherical domains due to limited clinical scanner resolutions and low SNR [1]. This paper presents a state-of-the-art dMRI deconvolution method to recover neuronal pathways in practical timeframes by developing highly efficient equivariant neural networks that account for dMRI's spatio-spherical data geometry and the voxel-level antipodal symmetry of neuronal fibers, all while demonstrating robustness to clinical dMRI resolutions.

**Need for deconvolution.** The diffusivity at a voxel reflects the underlying local tissue microstructure [7, 8, 44]. As multiple neuronal fibers can cross within a given resolution-limited voxel, recovering these fibers necessitates solving a blind fiber deconvolution problem at each voxel. Several voxel-level fiber models have been proposed [5, 45, 67, 69, 73, 79] and this work focuses on the widely used fiber Orientation Distribution Function (fODF) model, which represents the fiber configuration at a voxel as a sparse non-negative spherical function [66]. fODFs are the first step of common dMRI applications such as fiber tractography [37]. However, due to noise, subject motion, and clinically viable sampling resolutions (e.g., $\leq 30$ spherical samples per voxel), fODF recovery is highly ill-posed and requires significant regularization.

**Iterative solutions.** Constrained spherical deconvolution (CSD) [66] is the workhorse algorithm for dMRI deconvolution, solving the per-voxel fODF deconvolution problem iteratively, subject to non-negativity constraints. Several extensions of CSD further regularize fODFs to be sparse and/or spatially smooth [11, 12, 29, 57, 59, 74]. Further, as fODFs can be assumed to be antipodally symmetric [40, 67], many iterative algorithms optimize only over the hemisphere to accelerate

38th Conference on Neural Information Processing Systems (NeurIPS 2024).

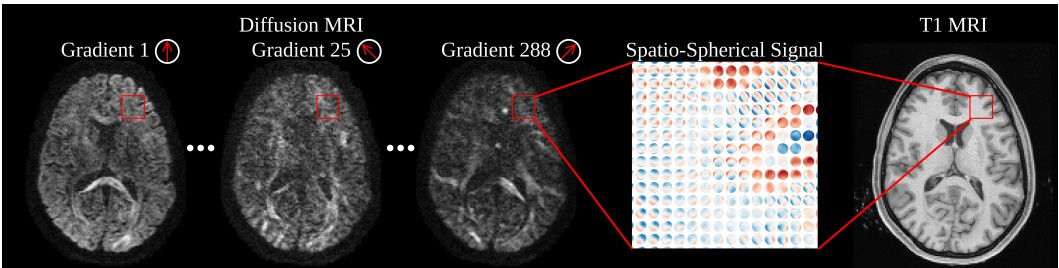

Figure 1: A diffusion MRI (columns 1–3) and a T1w MRI (column 5) derived from a subject in the HCP Young Adult dataset [70]. The inset (column 4) visualizes a region's spatio-spherical diffusion signal ($b - 1000mm/s^2$), highlighting crossing-fiber patterns and the grey/white matter interface.

per-voxel optimization. For example, CSD only optimizes for the even-order spherical harmonics of the fODF and RUMBA [11] uses approximately hemispherical sampling to represent the fODF. Despite significant progress using iterative solutions, these methods still require high angular sampling resolutions that are not clinically feasible to resolve crossing fibers within a voxel.

**Deep deconvolution networks.** Recent deep learning frameworks for dMRI deconvolution incorporate supervision and/or additional inductive biases to improve results. Supervised approaches train U-Nets to regress the solutions of iterative models or regress ground-truth fODFs estimated from ex vivo animal histology. These methods are consequently upper-bounded by the quality of iterative solutions and the generalizability limitations of small animal datasets. Further extensions incorporate the underlying spherical geometry of the per-voxel estimation by using $\mathbf{SO(3)}$-equivariant network layers but retain the supervised strategy of previous models [62]. ESD [19] instead proposes to use $\mathbf{SO(3)}$-equivariant network layers and *unsupervised* and regularized reconstruction-based deconvolution losses to surpass supervised solutions.

However, the above methods operate entirely at the voxel level and do not model the strong correlation between neighboring fODFs. To this end, RT-ESD [20] proposed $\mathbf{E(3)} \times \mathbf{SO(3)}$-equivariant network layers for dMRI that are equivariant to spatial rotations, translations, and reflections alongside voxel-wise spherical rotations to achieve high-quality deconvolution due to the joint spatio-spherical modeling. However, its high computational requirements limit its use in clinical or large-scale deployment. For instance, while subject-specific iterative fits with CSD require 3-5 minutes for the whole brain, RT-ESD requires about a day on an A100 GPU to converge when trained on a single subject.

**Contributions.** We present an efficient equivariant neural network framework for dMRI deconvolution that respects the data geometry of dMRI while ensuring computational practicality. Our spatio-hemispherical deconvolution (SHD) method addresses key redundancies and weaknesses of previous deep networks for dMRI deconvolution in three ways: (1) As fODFs are approximately antipodally symmetric at clinical resolutions, we replace RT-ESD's full spherical sampling with hemispherical sampling and find substantial efficiency gains by reducing the graph Laplacian of the voxel-wise SO(3) convolutions used in RT-ESD; (2) We then exploit the dense structure of dMRI data to further introduce optimized implementations and pre-computations, achieving a cumulative 65% reduction in processing time for an E(3) × SO(3)-equivariant U-Net; (3) Finally, we use explicit smoothness-promoting spatial regularization, leading to further improved fODF recovery.

Experimentally, we achieve state-of-the-art deconvolution results on two widely used simulated dMRI benchmarks with known ground truth. On real *in vivo* human dMRI, our method yields more spatially coherent fODF fields and higher robustness to changes in resolution from research-grade to clinical standards of single-shell low-angular protocols. Lastly, as the achieved efficiency gains enable training on a large set of human datasets, we can now train a single network for amortized inference on new human dMRI, as opposed to the subject-specific optimization of RT-ESD. Our code is available at `https://github.com/AxelElaldi/fast-equivariant-deconv`.

## 2 Related work

**Deep learning for dMRI.** Learned networks operating on dMRI have been extensively used to improve fiber tractography [46, 63], super-resolve diffusion signals or fODFs [48, 58, 78], regress

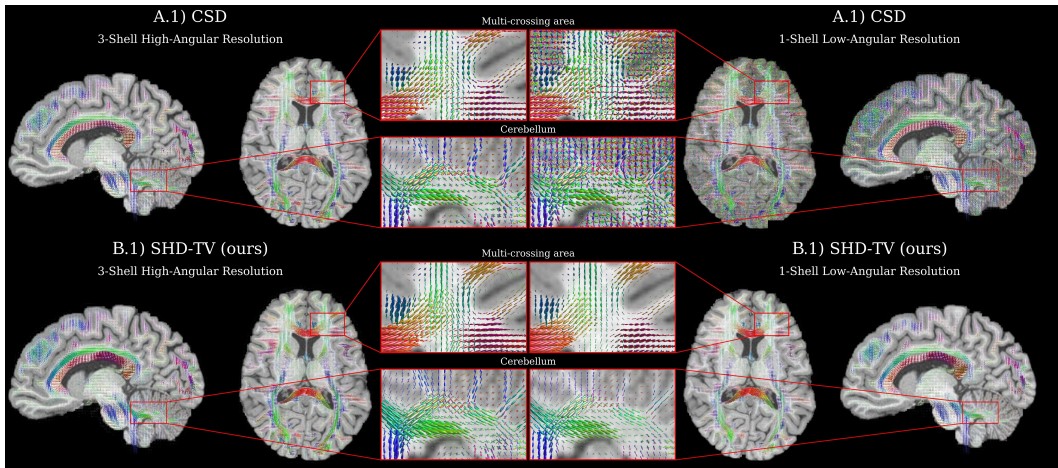

Figure 2: **A deconvolution visualization** comparing recovered fiber orientation distribution functions (fODFs) produced by the widely-used iterative CSD [66] model (**top row**) and our proposed SHD-TV model (**bottom row**) with high-resolution / clinically-infeasible (**left**) and low-resolution / clinically-feasible (**right**) spherical sampling. At high-resolutions (**left**), SHD-TV demonstrates enhanced localization of fiber orientations, heightened sensitivity to small-angle crossing fibers, and improved spatial consistency in the recovered fibers. At clinical low-resolutions (**right**), CSD struggles with the loss of input information, whereas our approach exhibits greater robustness to resolution losses and single-shell imaging protocols, yielding higher fidelity and spatially coherent fODFs. Appendix Fig. 9 visualizes comparisons with additional baselines.

microstructral indices in undersampled settings [13, 14, 27, 30, 31, 61], denoise artifacts [22, 23, 76], segment lesions [50], and more. Of these, most employ supervision via training on high-angular resolution research datasets to generalize to low-resolution clinical datasets. Our work is most related to networks that directly operate on $\mathbb{R}^3 \times \mathcal{S}^2$ images either through spherical networks applied voxel-wise or approximate parameterizations of the spatio-spherical space [6, 9, 10, 20, 19, 47, 50, 62].

For spherical data, $\mathbf{SO(3)}$-equivariant convolutions are often achieved via spherical harmonics-parameterized convolutions [16, 21, 42] or isotropic convolutions on spherical graphs [53]. These models have been extended to voxel-wise fODF estimation [19, 62] but do not use the spatial correlation between fODFs. [6, 10] develop convolutions for dMRI classification and super-resolution that exhibit voxel-wise $\mathbf{SO(3)}$-equivariance and incorporate manifold-valued spatial averaging. To model spatio-spherical dependence explicitly, $\mathbf{SE(3)}$-equivariance for improved dMRI segmentation has been achieved via tensor field networks [50] and separable kernels [9, 47] at the cost of high-memory usage. In contrast, RT-ESD [20] uses isotropic spatio-spherical kernels and focuses on deconvolution with equivariance to both joint and independent voxel and grid rotations. More recently, PONITA [9] proposes efficient $\mathbf{SE(3)}$-equivariant convolutions for spatio-spherical molecular graphs, but as yet, does not scale to high-angular resolutions needed for dMRI deconvolution.

**Spherical deconvolution methods.** Constrained spherical deconvolution (CSD) [38, 66] and its sparse [12] and/or spatially regularized extensions [11, 29, 57, 59, 74] are established but rely on lengthy and data-dependent iterative optimization and struggle to resolve small-angle crossing fibers in the low-resolution setting (see Fig. 2). More recently, several trainable models have been proposed for direct fODF estimation including pattern matching methods [24] that use a dictionary to match diffusion signal to known tissue microstructure and deep neural networks [19, 20, 39, 43, 46, 51, 52, 61, 62, 78] that learn a mapping from diffusion signals to fODFs. Trainable models have the advantage of potentially decreasing the need for high-angular resolution dMRI input. ESD [19] further introduced an unsupervised learning framework using only a diffusion signal reconstruction loss and additional fODF regularization, removing the need for ground-truth fODFs for training.

**Spatially informed deconvolution.** Traditionally, spherical deconvolution is optimized voxel-wise. However, the underlying tissue microstructure has long-range spatial correlations. Iterative methods account for this through spatial regularization such as total-variation [11] or fiber continuity [29, 57, 59, 74], at the cost of increased optimization complexity. Current neural network frameworks

have overlooked explicit fODF spatial regularization and rely solely on high-quality fODF training data and implicit spatial regularization by mean of spatial-weight sharing, either employing grid-wise 3D convolution [20, 46, 78], or channel-wise concatenation of neighboring voxels [19, 61, 62].

## 3 Methods

### 3.1 Background

**dMRI deconvolution.** A dMRI image is a spatio-spherical function $S : \mathbb{R}^3 \times \mathcal{S}^2 \to \mathbb{R}^B$ with $B$ features, called shells in the dMRI literature. The fODF model describes the tissue microstructure as a non-negative spatio-spherical function $F$, providing information on the local tissue composition and orientations, and links the dMRI and the fODFs $S = \mathcal{C}(F)$. dMRI deconvolution is interested in recovering the fODF from the dMRI by minimizing a constrained reconstruction loss $F^* = \text{argmin}_{F \geq 0} ||S - \mathcal{C}(F)||_2^2$. We focus our study on learnable spatio-spherical deconvolution operators trained to reconstruct high-angular resolution fODFs from sparse dMRI measurements $\mathcal{N}_\theta(S) = F$.

**Spatio-spherical convolutions.** $\mathbf{E}(3) \times \mathbf{SO}(3)$ is the group of independent $\mathbf{E}(3)$ and $\mathbf{SO}(3)$ transformations, where $\mathbf{E}(3)$ acts on the spatial grid $\mathbb{R}^3$ and $\mathbf{SO}(3)$ acts on the voxel sphere $\mathcal{S}^2$. Let $\psi_\theta : \mathbb{R}^3 \times \mathcal{S}^2 \to \mathbb{R}$ be a learnable filter, such that convolving $f$ and $\psi_\theta$ yields $f_{out}(T, R) = \int_{y \in \mathbb{R}^3} \int_{p \in \mathcal{S}^2} \psi_\theta(T^{-1}y, R^{-1}p) f(y, p) \, dp \, dy$ where $(T, R) \in \mathbf{E}(3) \times \mathbf{SO}(3)$. RT-ESD [20] implements these convolutions by only considering isotropic filters to limit computational complexity. In addition, the filter $\psi_\theta$ is expressed as a separable kernel $\psi_\theta(z_1, z_2) = \phi_{\theta_1}(z_1)\phi_{\theta_2}(z_2)$, turning the convolution into a two-step spherical-spatial convolution:

$$f_{\text{out}}(x, q) = \int_{y \in \mathbb{R}^3} \phi_{\theta_1}(||x - y||) \int_{p \in \mathcal{S}^2} \phi_{\theta_2}(qp^T) f(y, p) \, dp \, dy. \tag{1}$$

We build on RT-ESD [20], presented in Fig. 12, which uses DeepSphere [53]'s graph filtering based $\mathbf{SO}(3)$-equivariant spherical convolution. At every spatial location $y \in \mathbb{R}^3$, $f(y, .)$ is discretized on a set of spherical vertices $\mathcal{V} = \{q_i \in \mathcal{S}^2\}_{i \in [1,..,N_\mathcal{V}]}$, $\mathbf{f}(y) \in \mathbb{R}^{N_\mathcal{V}}$. The graph $\mathcal{G} = (\mathcal{V}, \mathbf{A})$ and its normalized Laplacian $\mathbf{L}$ are constructed from the set $\mathcal{V}$, where $\mathbf{A} \in \mathbb{R}^{N_\mathcal{V} \times N_\mathcal{V}}$ is the spherical adjacency matrix. The voxel-wise spherical filtering output is then used to construct a spatio-spherical graph $\hat{\mathbf{f}}(y) = [T^0(\mathbf{L})\mathbf{f}(y), ..., T^{K-1}(\mathbf{L})\mathbf{f}(y)] \in \mathbb{R}^{N_\mathcal{V} \times K}$ with $K$ features, where $K$ is the polynomial degree of the filtering and $T^k(\mathbf{L})$ is the Chebyshev polynomial of degree $k$. A learnable isotropic spatial filter $\alpha_k : \mathbb{R}^+ \to \mathbb{R}$ is then defined for each spherical filtered map $\hat{\mathbf{f}}_k$, resulting in the overall $\mathbf{E}(3) \times \mathbf{SO}(3)$-convolutional operation expressed as:

$$\mathbf{f}_{out}(x) = \sum_{y \in \mathcal{P}(x)} \sum_{k=0}^{K-1} \alpha_k(||x - y||) T^k(\mathbf{L})\mathbf{f}(y), \tag{2}$$

where $\mathcal{P}(x)$ is a spatial neighborhood around the position $x \in \mathbb{R}^3$. Spherical $\mathbf{SO}(3)$-equivariance is achieved by using a uniform and symmetric $\mathcal{V}$ sampling such as the HEALPix [32] grid, providing a uniform coverage of the spherical domain at different levels of resolution, and graph weight $p, q \in \mathcal{V}$, $\mathbf{A}(p, q) = e^{-||p-q||/\sigma}$ with $\sigma$'s value selected to minimize empirical equivariance error.

### 3.2 *Efficient* dMRI spatio-hemispherical convolutions

**Spatio-Hemispherical Equivariant convolution.** fODFs and dMRIs are approximately antipodally symmetric, which makes fully spherical convolutions redundant. We therefore improve the time and space efficiency of the RT-ESD convolution by reducing the spherical graph $\mathcal{G}$ to a hemispherical graph $\mathcal{H}$. Because $f$ is antipodal symmetric, we assume, without loss of generality, that $\mathcal{V}$ is a symmetric sampling, i.e. $\forall p \in \mathcal{V}, \exists (\text{-}p) \in \mathcal{V}$. We construct the hemispherical sampling $\mathcal{V}^+ = \{p \in \mathcal{V} \text{ and } p_z \geq 0\}$ from the spherical sampling $\mathcal{V}$, with $p_z$ chosen to be the 3rd spatial coordinate of $p$. We compute the hemispherical Laplacian weight as,

$$p, q \in \mathcal{V}^+, \ \mathbf{L}^+(p, q) = \mathbf{L}(p, q) + \mathbf{L}(p, -q). \tag{3}$$

Thus, the spherical operation $\mathbf{L}f$ is entirely explained by the hemisphere graph Laplacian $\mathbf{L}^+ f^+$, i.e. $\forall p \in \mathcal{V}, \exists (p^+) \in \mathcal{V}^+$ s.t. $(\mathbf{L}^+ f)(p^+) = (\mathbf{L}f)(p)$, where $f^+$ is the $f$ sampling on $\mathcal{V}^+$ (proof

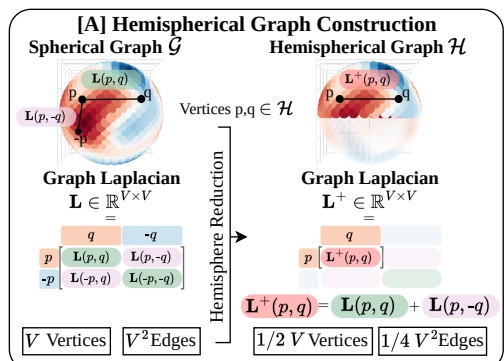 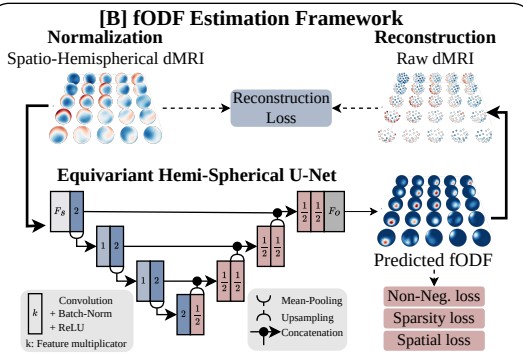

Figure 3: **Contribution overview. A.** We reduce the spherical graph $(\mathcal{G}, \mathbf{L})$ to an hemispherical graph $(\mathcal{H}, \mathbf{L}^+)$. **B.** The SHD deconvolution framework operates on a grid of spherical signals and reduces computation complexity while improving neuronal fiber deconvolution.

in appendix A.1). For spherical samplings $\mathcal{V}$ such as HEALPix, our proposed sampling reduces the number of vertices sampled to $\frac{1}{2}|\mathcal{V}|$ and the hemispherical Laplacian size to $\frac{1}{4}|\mathcal{V}|^2$. Our reduction is both theoretically and empirically equivariant to $\mathbf{E(3)} \times \mathbf{SO(3)}$, see appendix A.2 for more details.

**Dense Matrix Multiplication.** DeepSphere's [53] used sparse matrix multiplication to compute $T^k(\mathbf{L})\mathbf{f}(x)$. However, for dMRI, our Laplacians are dense and sparse matrix multiplication adds significant computational overhead on dense matrices. To address this, we substitute the sparse matrix multiplication used in [53] with standard matrix multiplications and find substantial efficiency improvements when applied to dMRI data.

**Pre-computed Chebyshev Polynomials.** DeepSphere [53] assumes the spherical sampling $\mathcal{V}$, and thus the Laplacian $\mathbf{L}$, to be dependent on the input spherical signal $\mathbf{f}$. Thus, for efficiency, the Chebyshev polynomials are computed iteratively as $T^{k+1}(\mathbf{L})\mathbf{f}(x) = (2\mathbf{L}T^k(\mathbf{L}) - T^{k+1}(\mathbf{L}))\mathbf{f}(x)$ for each new $\mathbf{f}$ and $\mathbf{L}$. In our setting, we use the same spherical sampling $\mathcal{V}$ for every spherical signal $\mathbf{f}$, making the Laplacian $\mathbf{L}$ of every convolutional layer fixed. We therefore precompute and store these polynomials before training to further eliminate redundancy.

### 3.3 Spatial Hemispherical Deconvolution (SHD) Network

Fig.3 presents an overview of our proposed spatial-hemispherical deconvolution (SHD) framework. Below, we first describe how our inputs are preprocessed and then detail our network and the losses used to train it.

**Data normalization.** The dMRI signal acquisition yields a raw spatio-spherical signal $\mathbf{S}_{\mathcal{G}}$ sampled on a set of spherical coordinates $\mathcal{G}$, called shell sampling. The signal intensity range and shell sampling are scan-dependent and need to be normalized before being fed to our proposed neural network. Following [55], we normalize the dMRI scans such that the white matter $B0$-diffusion signal is comparable between scans. Then, following [19], the normalized data is interpolated onto a hemispherical HEALPix sampling $\mathcal{V}^+$. We provide details in appendix C.2.

$\mathbf{E(3)} \times \mathbf{SO(3)}$**-Unet.** The interpolated diffusion signal is then deconvolved to recover the spatio-spherical fODFs $\mathbf{F}_{\mathcal{V}+}$ on the same input HEALPix sampling $\mathcal{V}^+$. We use a U-Net, with 4 depth levels, with layers acting on a spatial grid of hemispherical samples. We use our proposed layers in the U-Net architecture presented in Fig. 3. Besides the last, every block comprises a convolution, batch-norm, and a ReLU activation. The last block consists of a convolution and a Softplus activation function. The up/downsampling layers first involve mean spatial upsampling/pooling on $\mathbb{R}^3$, followed by mean spherical upsampling/pooling on $\mathcal{S}^2$, both introducing minor numerical equivariance error. Following DeepSphere [53], to minimize the equivariance error from the spherical upsampling/pooling, we use the hierarchical structure of the HEALPix grid.

**Network training.** Our proposed deconvolution network $\mathcal{N}_\theta$ takes as input the shell-sampled dMRI signal $\mathbf{S}_{\mathcal{G}}$ to produce fODFs $\mathbf{F}_{\mathcal{V}+}^\theta = \mathcal{N}_\theta(\mathbf{S}_{\mathcal{G}})$. We train the network in an unsupervised manner by reconstructing the dMRI signal $\mathbf{S}_{\mathcal{D}}^\theta = \mathcal{C}(\mathbf{F}_{\mathcal{V}+}^\theta)$ from the estimated fODFs, with details on the

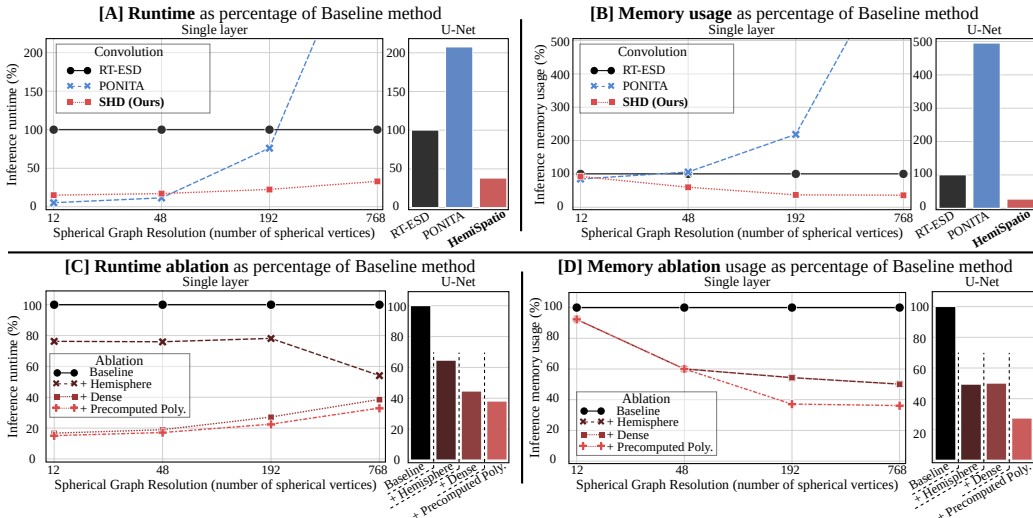

Figure 4: **Efficiency analysis.** Runtime (**A** & **C**) and GPU memory usage (**B** & **D**) expressed as the percentage of the baseline [20], for both: (**line plots**) a convolutional layer applied to increasing angular resolution samplings and (**bar plots**) a U-Net applied to high-angular resolution. The proposed convolution is more efficient than existing equivariant spatio-spherical convolutions.

reconstruction function $\mathcal{C}$ in appendix C.2. We note that the input and reconstruction shell samplings $\mathcal{G}$ and $\mathcal{D}$ need not be the same, allowing for angular super-resolution reconstruction during training if required. The parameters $\theta$ are estimated by minimizing a reconstruction loss subject to non-negativity [19, 66] and sparsity-promoting regularization [12, 19]:

$$\mathcal{L} = ||\mathbf{S}_{\mathcal{D}} - \mathbf{S}_{\mathcal{D}}^{\theta}||_2^2 + \lambda_{\text{nn.}}||\mathbf{A}.\mathbf{F}_{\mathcal{V}+}^{\theta}||_2^2 + \lambda_{\text{spa.}}||\log\left(1 + \frac{\mathbf{F}_{\mathcal{V}+}^{\theta}}{\sigma^2}\right)||_2^2 + \lambda_{\text{tv}}\mathcal{L}_{\text{tv}}, \qquad (4)$$

where the terms correspond to reconstruction, non-negativity, sparsity, and smoothness, respectively. We define $\mathbf{A}$ as a vector setting any positive component of $\mathbf{F}_{\mathcal{V}+}^{\theta}$ to 0. We set $\sigma = 10^{-5}$ following [12]. $\mathcal{L}_{TV}$ is described below.

**Spatial regularization.** To add *explicit* spatial regularization, we propose to extend the spatial total-variation regularization from [11] to the trainable model framework $\mathcal{L}_{\text{tv}} = ||\nabla_x \mathbf{F}_{\mathcal{V}+}^{\theta}||_2^2$ where $\nabla_x$ is the spatial gradient operator. The spatial regularizer $\mathcal{L}_{\text{tv}}$ seeks to promote smoother reconstructions in ill-posed settings, especially at clinical angular resolutions used later in our experiments.

## 4 Experiments

We first quantify the runtime and memory efficiency gains produced by our contributions. We then analyze their use across a diversity of dMRI deconvolution settings on both real and synthetic datasets.

### 4.1 Runtime and memory efficiency improvements

**Experimental details.** We first compare the efficiency of our proposed convolution against two spatio-spherical equivariant convolutions: RT-ESD [20] and PONITA [9], as measured by inference runtime and memory consumption. We then conduct an ablation analysis of our three proposed improvements. All analyses are conducted for both a single spatio-spherical layer applied to increasing spherical graph resolution and also for an entire spatio-spherical U-Net architecture detailed in Fig. 3. As input, we use a random $8 \times 8 \times 8 \times V$ spatio-spherical volume, with $V$ depending on the spherical HEALPix grid resolution. We fix the U-Net input HEALPix resolution to 8 ($V = 754$) and we vary the single layer HEALPix resolution from 1 ($V = 12$) to 8. We take the RT-ESD convolutions as a reference baseline and present runtime and memory usage results as a percentage of the baseline averaged across 50 inference steps. All comparisons are performed using an Nvidia RTX-8000 GPU.

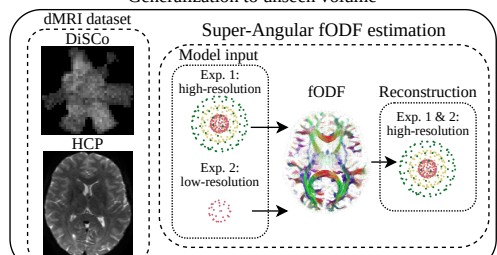
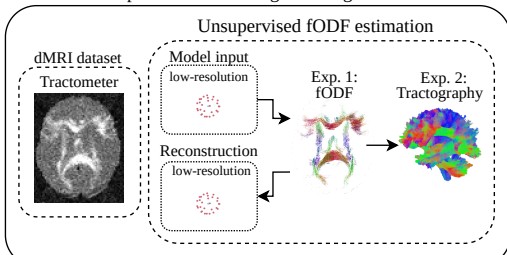

Figure 5: **Overview of the diffusion MRI experiments in Section 4.2**. **[A]** We perform super-resolved fODF estimation experiments on two datasets, DiSCo and HCP, respectively. Here, we study the impact of using either high-angular or low-angular resolution as input. **[B]** We perform quantitative fODF and tractography estimation experiments on Tractometer. We extract fODFs and tractograms from the dMRI with both input and output having low-angular resolution.

**Results.** Fig. 4 presents the results of the efficiency analysis. For a single layer, our convolutions are 2 to 5 times faster than the RT-ESD baseline while decreasing memory footprint by 2.5 times on large spherical sampling. PONITA has a similar runtime and memory footprint as SHD on sparse spherical sampling but underperforms on large spherical graphs (that are common in dMRI deconvolution [12]), being 10 times slower and more memory intensive. Consequently, a U-Net designed for high-angular resolution sampling is 2.5 and 3.5 faster using our proposed convolutions over RT-ESD and PONITA, while using 4 and 20 times less memory, respectively. Our ablations indicate that using our hemispherical convolutions over the spherical convolutions, as well as using precomputed Laplacian polynomials, brings the most improvement for both runtime and memory on large spherical sampling while leveraging dense matrix multiplication reduces runtime on low-angular resolutions.

## 4.2 Diffusion MRI deconvolution experiments

**Experimental details.** Validation of fODF estimation methods is confounded by the absence of known and unbiased ground truth fODFs in *in vivo* human dMRI. To address this limitation, the literature commonly benchmarks methods on synthetic datasets with known ground truth tissue microstructure. We conduct both a quantitative (synthetic dataset) and qualitative (*in vivo* human dMRI) analysis of our deconvolution framework (Section 4.2.1). We then perform a quantitative analysis on the tractography downstream task on a benchmark synthetic dataset (Section 4.2.2). For quantitative evaluation, we first estimate fODFs, then detect fiber peaks, and report the peak angular error and false positive rate (FPR) following [17]. The angular error is the average angle between the ground truth peaks and their closest predicted peaks and the FPR is the number of predicted peaks that do not match any ground-truth peaks divided by the number of ground-truth peaks. See Appendix A.3 for more details. All deep network experiments were performed using a single RTX8000 GPU and used less than 16GB of system RAM.

**Baselines.** We compare the proposed SHD method against conventional iterative methods including CSD [66], RUMBA and RUMBA-TV [11], and state-of-the-art learning based methods leveraging different convolutional layer, such as the non-equivariant voxel-wise MLP in [52] and the volume-wise CNN in [46], that we also extend to have spatial regularization (CNN-TV). We also compare against equivariant voxel-wise spherical convolutions ESD [61, 19], and more recent equivariant spatio-spherical convolutions RT-ESD [20] and PONITA [9]. To focus on the inductive bias introduced by each method's convolution layer, we adopt a standardized network architecture and training framework, presented in section 3.3. We provide details about these different baselines in appendix C.3.

**Data.** We leverage three public dMRI datasets for validation. DiSCo [54] is a synthetic dMRI dataset with three volumes, split into training, validation, and testing volumes, with high-angular resolution sampling (4 shells each with 90 gradients) provided with ground truth fODF. We then use 100 in-vivo unrelated dMRI scans from the HCP young adult dataset [70], split into 65 training, 15 validation, and 20 testing volumes, with high-angular resolution (3 shells, each with 90 gradients), without any ground truth. As we aim to benchmark performance on both high-angular resolution data and more commonly acquired low-angular resolution data, we further simulate low-angular

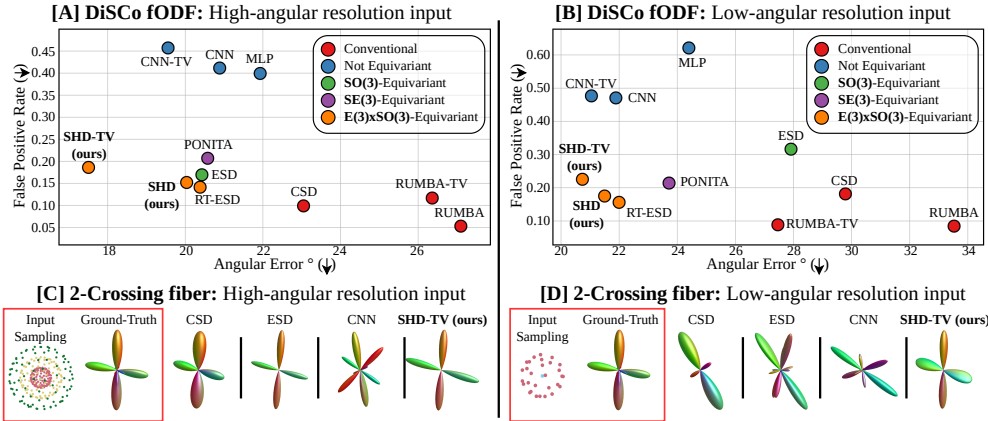

Figure 6: **DiSCo fiber detection performances** on high (**left col.**) and low (**right col.**) angular resolutions. We first present fODF estimation results on high-angular **[A]** and low-angular resolution **[B]** input (**closer to bottom-left** is better). **[C-D]** then present a qualitative example of a two-crossing fiber estimation. Our faster implementation SHD does not negatively impact results in comparison to RT-ESD, while our improved model SHD-TV outperforms other methods by providing higher angular precision and less spurious fibers, especially at clinically-viable low-angular resolution.

resolution acquisitions on DiSCo and HCP by randomly selecting 29 gradients from only the lowest shell. The last dataset is Tractometer [49], a single-volume simulation of a real human brain with a low-angular resolution protocol (1 shell, 32 gradients), provided with ground truth tractography alongside a tractography scoring algorithm [60].

### 4.2.1 Super-resolved fODF estimation: synthetic DiSCo and real human HCP datasets

We perform two experiments, depicted in Fig.5A). First, we measure performance when all baselines are trained and tested on high-angular resolution data. Second, we give all methods high-angular resolution data only during training and test them on low-angular resolution data. In this setting, the networks are trained to regress the high-angular resolution data from low-angular resolution input. For quantitative evaluation, we extract fiber directions from the estimated fODF and we report the peak angular error and false positive rate as described in the previous section. All results are averaged over five random training seeds. Finally, as HCP lacks fODF and tractogram ground truth, we perform a qualitative analysis. Further validation details are in appendix A.3 and A.4.

**Results.** Fig. 6 presents DiSCo deconvolution results for both the low and high-angular resolution settings. We visualize qualitative results on a random HCP test subject in Fig.2, focusing on two areas with diverse tissue microstructures, such as crossing fibers, fiber bending, and multi-tissue compartments. Our proposed efficiency contributions retain all of the performance of previous methods as our proposed SHD model (without spatial regularization) achieves similar results to the much slower and more memory-intensive RT-ESD and PONITA methods. Moreover, non-equivariant methods, while having competitive angular errors, are negatively impacted by a high false positive rate and qualitatively high number of spurious fibers.

Finally, on research-grade high-angular resolution data, incorporating spatial information in the network input does not improve results. However, adding total-variation regularization on the output of the model as in our proposed SHD-TV model shows significantly improved results. Further, in a clinical low-angular setting, our spatially informed models (SHD and SHD-TV) widely outperform voxel-wise methods, showcasing the importance of contextual information for clinical dMRI.

### 4.2.2 Unsupervised fODF estimation and tractography w/ known ground truth: Tractometer

A high-level overview of this experiment is described in Fig.5B). As Tractometer consists of a single volume with a low-angular resolution protocol with no held-out testing data, we benchmark methods on unsupervised fODF estimation similarly to previous work [19]. We first train the models in an

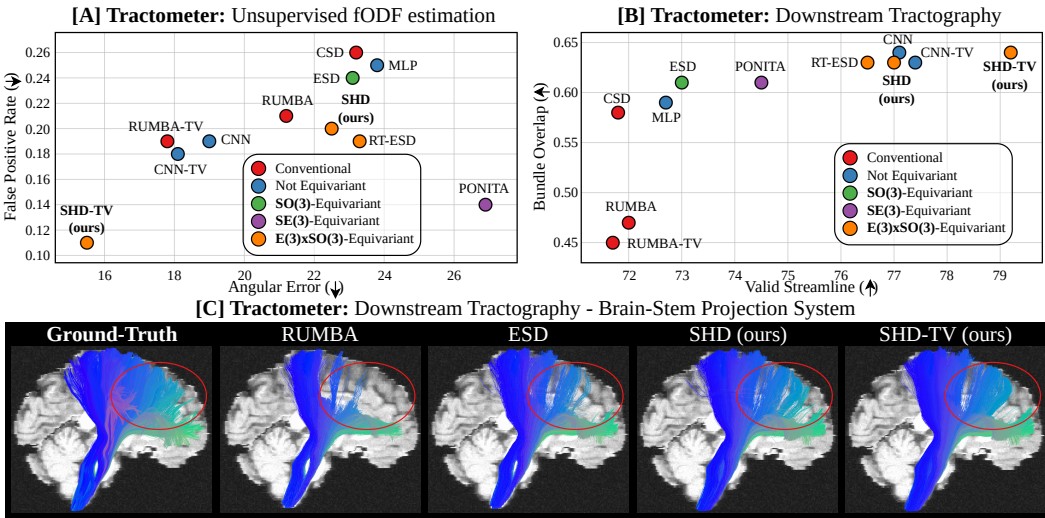

Figure 7: **Tractometer fODF estimation and tractography performance. Top:** Unsupervised fODF estimation (**A**, closer to bottom left is better) and tractography (**B**, closer to top right is better) results. **Bottom:** In **[C]**, we visualize ground-truth and estimated fibers projecting out from the brainstem into the right hemisphere. Overall, `SHD` and `SHD-TV` demonstrate more faithful fiber and streamline recovery as compared to the voxel-wise `RUMBA` and `ESD` methods. In particular, `SHD-TV` yields fewer invalid streamlines and increases spatial coherence.

unsupervised manner to estimate fODFs and then use them to investigate the effect of improved local fODF estimation on white matter tractography accuracy. Validation of the unsupervised estimated fODFs involves using the same local fiber detection evaluation as for the DiSCo experiments. Subsequently, the validation of the estimated tractographs is conducted by comparing them against 21 ground truth fiber bundles provided by the dataset [60]. The white matter streamlines of the tractograph are initially segmented into 21 target fiber bundles. The correctly identified bundles are counted as Valid Bundles and the streamlines belonging to one of the ground truth bundles are counted as Valid Streamlines. Further, the volume of each estimated bundle is compared against the volume of its corresponding ground truth bundle, leading to the computation of the Overlap score (analogous to a True Positive value) and the Overreach Score (similar to a False Positive value).

**Results.** We present Tractometer results in Fig.7. In this setting where all methods are trained and tested on the same volume, non-equivariant methods (`CNN` and `CNN-TV`) outperform most unregularized equivariant deconvolution methods. We speculate that when generalization is not required, non-equivariant models enjoy a higher overfitting capability in this problem, thereby diminishing the advantage of employing inductive priors in neural networks.

However, the proposed `SHD-TV`, which incorporates both spatio-spherical inductive biases and spatial regularization, significantly outperforms all other methods, regardless of whether they are equivariant or spatially regularized. This finding supports the compounding benefits of both our architectural modifications and our proposed regularization strategy. In turn, these enhanced fODF recovery performances positively impact tractography accuracy, reflected in a quantitatively higher number of valid streamlines and bundle overlap (albeit with an increase in bundle overreach) and more qualitatively coherent streamlines.

## 5    Discussion

**Limitations and future work.** Our work has certain limitations. First, while we focus on the key dMRI task of fiber deconvolution, our proposed layers are task-agnostic and can also be deployed to other tasks such as denoising and segmentation in future studies. Second, while our layers can be used in many fields with spatio-spherical data outside of neuroimaging, such as robotics [72], neural rendering [26], and molecular dynamics [9], our assumption of antipodal symmetry (widespread in dMRI) may need to be relaxed for many applications within these fields. Third, angularly

undersampled reconstruction results have an inherent risk of missing fibers or fiber hallucination. While our work partially mitigates these unrealistic reconstructions by adding geometric and spatial priors to the reconstruction, future validation studies could analyze cohorts with paired scans on clinical low-resolution and research-grade high-resolution dMRI protocols. Finally, our methods need further testing on additional clinical challenges such as subject motion, imaging artifacts, and abnormal structures. However, as our work is entirely self-supervised and only based on regularized reconstruction of input data, we anticipate robustness to such imaging variation. Preliminary results on a pathological brain sample are provided in appendix A.5.

**Conclusions.** The analysis of *in vivo* white matter neuronal organization within the brain depends on the key task of dMRI deconvolution and current methods are either not accurate or too slow. To that end, we developed highly time and memory-efficient equivariant convolutional network layers that respect the physical symmetries of dMRI. Our contributions led to $3.5\times$ faster runtimes and up to $20\times$ lower memory consumption as compared to existing spatio-spherical layers, while exceeding or maintaining their performance on dMRI studies. These layers were then used to construct networks that achieved strong dMRI deconvolution performance in practical timeframes across multiple commonly used synthetic benchmark datasets and real *in vivo* human datasets. Finally, our methods are robust to the dMRI angular resolutions typically used in clinical practice outside of research settings, potentially enabling robust analyses of large-scale retrospective dMRI studies and direct application to the clinic.

## Acknowledgements

The authors gratefully acknowledge support from NIH grants 1R01HD088125-01A1, R01HD055741, 1R01MH118362-01, 2R01EB021391-06 A1, R01ES032294, R01MH122447, and 1R34DA050287.

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

# A  Additional results

## A.1  Spherical and hemisphere convolution

**Hemispherical sampling.** Consider the symmetric spherical sampling $\mathcal{V} = \{p \in \mathbb{S}^2\}$ such that $\forall p \in \mathcal{V}, \exists (-p) \in \mathcal{V}$, and the Euclidean coordinate $p = (p_x, p_y, p_z) \in \mathcal{S}^2$. The sphere can be divided into a north and south hemisphere $\mathcal{V} = \mathcal{V}^+ \cup \mathcal{V}^-$, with $\mathcal{V}^+ = \{p \in \mathcal{V}, \text{ such that } p_z > 0)\}$ and $\mathcal{V}^- = \{p \in \mathcal{V}, \text{ such that } p_z < 0)\}$. For point $p$ lying on the circle $p_z = 0$, we further add to $\mathcal{V}^+$ every point such that $p_y > 0$ and to $\mathcal{V}^-$ every point such that $p_y < 0$. Finally, for the two points $p$ such that $p_z = p_y = 0$, we add $p$ such that $p_x > 0$ to $\mathcal{V}^+$ and $p$ such that $p_x < 0$ to $\mathcal{V}^-$.

**Hemispherical restriction.** Consider a spherical function $f : \mathcal{S}^2 \to \mathbb{R}$ and a function $\mathbf{L} : \mathcal{S}^2 \times \mathcal{S}^2 \to \mathbb{R}$, both sampled on $\mathcal{V}$ such that $f \in \mathbb{R}^V$ and $\mathbf{L} \in \mathbb{R}^{V \times V}$. Consider the function $\mathbf{L}^+ : \mathcal{S}^2 \times \mathcal{S}^2 \to \mathbb{R}$, $\mathbf{L}^+(p, q) = \mathbf{L}(p, q) + \mathbf{L}(p, -q)$, sampled on $\mathcal{V}^+$ such that $\mathbf{L}^+ \in \mathbb{R}^{V/2 \times V/2}$ and $f^+ \in \mathbb{R}^{V/2}$ the sampling of $f$ on $\mathcal{V}^+$.

**Theorem 1** *Consider an antipodally symmetric spherical function $f$ and $\mathbf{L}$, such that for all $p, q \in \mathcal{V}$, $f(p) = f(-p)$ and $\mathbf{L}(p, q) = \mathbf{L}(\text{-}p, \text{-}q)$, then $\forall p \in \mathcal{V}, \exists p^+ \in \mathcal{V}^+$, such that $(\mathbf{L}f)(p) = (\mathbf{L}^+ f^+)(p^+)$.*

**Proof** Let $p \in \mathcal{V}$. Then, define $p^+ = p$ if $p \in \mathcal{V}^+$, -$p$ otherwise. Thus, we know that $p^+ \in \mathcal{V}^+$.

$$(\mathbf{L}f)(p) = \sum_{q \in \mathcal{V}} \mathbf{L}(p, q) f(q) \tag{5}$$

$$= \sum_{q \in \mathcal{V}^+} \mathbf{L}(p, q) f(q) + \sum_{q \in \mathcal{V}^-} \mathbf{L}(p, q) f(q) \tag{6}$$

$$= \sum_{q \in \mathcal{V}^+} \mathbf{L}(p, q) f(q) + \mathbf{L}(p, -q) f(-q) \tag{7}$$

$$= \sum_{q \in \mathcal{V}^+} (\mathbf{L}(p, q) + \mathbf{L}(p, -q)) f(q) \tag{8}$$

$$= \sum_{q \in \mathcal{V}^+} (\mathbf{L}(p^+, q) + \mathbf{L}(p^+, -q)) f(q) \tag{9}$$

$$= \sum_{q \in \mathcal{V}^+} \mathbf{L}(p^+, q) f(q) \tag{10}$$

$$= (\mathbf{L}^+ f^+)(p^+) \tag{11}$$

$$\tag{12}$$

## A.2  Numerical equivariance error analysis

**Motivation.** Due to discretization and implementation practicalities, equivariant networks are subject to aliasing and are only approximately equivariant [33]. Prior work in this space [53, 19, 20] has demonstrated low equivariance errors for $\mathbf{SO(3)}$ and $\mathbf{E(3)} \times \mathbf{SO(3)}$ convolutions and their U-Net counterparts when operating on full spherical sampling. We claim that the proposed accelerated implementations of these convolutions do not introduce additional equivariance error and provide empirical evidence below.

**Model details.** The hemispherical $\mathbf{E(3)} \times \mathbf{SO(3)}$ convolution is compared to a grid-wise $\mathbf{E(3)}$-equivariant convolution with spherical information, $\mathbf{E(3)}$-SH, and a voxel-wise $\mathbf{SO(3)}$-equivariant convolution with spatial information, Concat-$\mathbf{SO(3)}$. The $\mathbf{E(3)}$-SH convolution is a 3D convolution with isotropic kernels processing spherical harmonic coefficients as input features [52]. The Concat-$\mathbf{SO(3)}$ convolution is a spherical convolution processing spatial neighbors as input features [62, 20]. The spatial and spherical kernels are initialized randomly following [34].

**Evaluation.** We measure numerical equivariance error as the deviation from the equivariance equality, using:

$$\nabla \text{Equiv}(\mathcal{N}, G, f) = \frac{||[G \mathcal{N}(f)] - \mathcal{N}([Gf])||_2^2}{||\mathcal{N}(f)||_2^2} \tag{13}$$

where $\mathcal{N}$ is the tested operator, $f$ is a spatio-spherical signal, and $G = (T, R) \in \mathbf{E}(\mathbf{3}) \times \mathbf{SO}(\mathbf{3})$ is a composition of grid-wise and voxel-wise transformation. In practice we only test rotation-equivariance, and we limit $T$ and $R$ to be rotations. This quantity is measured over $1000$ randomly sampled spatio-spherical signals $f$, random rotations $(T, R)$, and randomly initialized $\mathcal{N}$. $f$ is randomly generated from a uniform $\mathcal{U}[0, 1]$ distribution on a $8 \times 8 \times 8 \times V$ spatio-spherical volume with HEALPix hemispherical grid of resolution $8$ ($V = 384$ vectors). Moreover, we filter high spherical frequency by only keeping the first $8$ spherical harmonic degrees.

**Results.** Fig.8 illustrates estimated equivariance errors. As expected, we find that the $\mathbf{E}(\mathbf{3})$-SH convolution exhibits a lack of equivariance to voxel-wise rotations, and the Concat-$\mathbf{SO}(\mathbf{3})$ convolution exhibits a lack of equivariance to grid-wise rotations. The accelerated implementation of the $\mathbf{E}(\mathbf{3}) \times \mathbf{SO}(\mathbf{3})$ convolution, taking into account both inductive bias of the spatial and spherical domain, maintains low voxel-wise rotation equivariance errors.

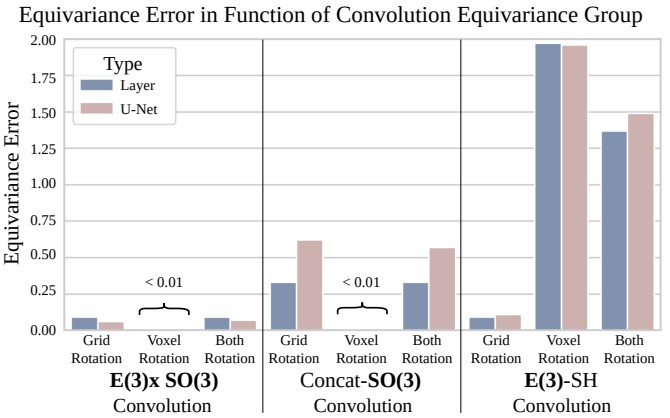

Figure 8: Quantitative evaluation of equivariance error, depending on the convolution equivariance group, and the applied rotation group.

### A.3 DiSCo quantitative results

**Experiment details.** We provide more details on the DiSCo dataset, which we used to evaluate both fODF estimation performance and robustness to dMRI protocol angular resolution. The DiSCo dataset comprises three $40 \times 40 \times 40$ synthetic volumes with variable amounts of synthetic noise added. Our experiments use the volumes with a noise level of SNR=30. All three volumes share the same synthetic generation process and have four B0 images and four shells each with $90$ gradients. We do not apply any pre-processing step. For quantitative analysis, we train the different models on the first volume, validate on the second, and test on the third. Given that the synthetic generation process of the DiSCo dataset is similar to a white matter/CSF tissue simulation, we limit the different models to a 2 tissue decomposition, and we analyze the white-matter fODFs.

**Validation details.** All results for deep network models are averaged over five random seeds. For quantitative evaluation, we first estimate fODFs, then detect peaks, and finally threshold them based on their voxel-wise relative fiber partial volume applying 19 different threshold values ranging from $0.05$ to $0.95$. For each threshold, we calculate the false positive rate (FP), false negative rate (FN), and angular precision of detected fibers following the evaluation framework proposed by [17]; utilizing a rejection cone of $25°$. Precision and recall are computed at different thresholds using FP, FN, and true positive fibers (TP). Subsequently, we calculate the Precision-Recall F1 score. We report the angular error and false positive rate for the threshold that maximizes the F1 score on the DiSCo validation volume. We report raw average scores and $95\%$ confidence interval (CI) average over five random seeds in Table 1. Notice that all scores have a CI$< 0.01$, but the angular error.

### A.4 Qualitative in vivo human results

**Experiment details.** We provide more details on the HCP dataset, which we used for qualitative evaluation of the fODF estimation. The HCP young adult release contains $100$ unrelated subjects and

Table 1: DiSCo fiber detection performances @ noise level SNR=30 on high and low-resolution data. Models requiring training are trained on the first volume, validated on the second, and tested on the third. Results average over 5 random initialized models. Confidence interval at 95% given if CI is greater than 0.01.

| Model | PR AUC (↑) | | F1 score (↑) | | Angle @F1 (↓) | | FNR @F1 (↓) | | FPR @F1 (↓) | |
|---|---|---|---|---|---|---|---|---|---|---|
| Protocol resolution | High | Low | High | Low | High | Low | High | Low | High | Low |
| CSD | 0.64 | 0.50 | 0.56 | 0.45 | 23.0 | 29.8 | 0.65 | 0.74 | 0.10 | 0.18 |
| RUMBA | 0.61 | 0.54 | 0.49 | 0.42 | 27.1 | 33.5 | 0.74 | 0.80 | **0.05** | 0.08 |
| RUMBA-TV | 0.49 | 0.55 | 0.49 | 0.48 | 26.4 | 27.5 | 0.71 | 0.74 | 0.12 | 0.09 |
| MLP | 0.59 | 0.48 | 0.53 | 0.42 | $23.8 \pm 0.1$ | $31.8 \pm 0.2$ | 0.68 | 0.78 | 0.12 | 0.15 |
| CNN | 0.56 | 0.51 | 0.51 | 0.47 | $24.9 \pm 0.3$ | $27.4 \pm 0.3$ | 0.70 | 0.74 | 0.11 | 0.12 |
| $S^2$ U-net | 0.63 | 0.50 | 0.58 | 0.42 | $22.2 \pm 0.1$ | $32.4 \pm 0.1$ | 0.62 | 0.79 | 0.11 | 0.13 |
| ESD | 0.64 | 0.47 | 0.61 | 0.47 | $20.4 \pm 0.2$ | $27.9 \pm 0.8$ | 0.56 | 0.68 | 0.17 | 0.32 |
| Concat-ESD | 0.65 | 0.59 | 0.61 | 0.56 | $20.3 \pm 0.1$ | $22.6 \pm 0.1$ | 0.56 | 0.61 | 0.15 | 0.20 |
| RT-ESD | 0.64 | 0.60 | 0.60 | 0.57 | $20.4 \pm 0.3$ | $22.0 \pm 0.2$ | 0.58 | 0.61 | 0.14 | 0.16 |
| SHD (ours) | 0.64 | 0.61 | 0.60 | 0.57 | $20.0 \pm 0.1$ | $21.5 \pm 0.5$ | 0.57 | 0.60 | 0.17 | 0.19 |
| SHD-TV (ours) | **0.70** | **0.62** | **0.65** | **0.58** | $\mathbf{17.5 \pm 0.2}$ | $\mathbf{20.7 \pm 0.2}$ | **0.49** | 0.57 | 0.19 | 0.22 |

ensures the exclusion of twin subjects to prevent any data leakage between dataset splits. The dMRI volumes had undergone pre-processing using FSL [36] and FreeSurfer [25], and further details on the preprocessing pipeline can be found in the HCP dataset documentation [28, 2, 3, 4]. The 100 dMRI volumes adhered to a high-resolution diffusion protocol, featuring 18 B0 volumes per subject and three diffusion shells, each comprising 90 diffusion gradients. We apply similar downsampling to the HCP dataset, randomly selecting 29 diffusion gradients from the lower b-value shell $(1000, s/mm^2)$ while retaining the B0 volumes. For qualitative analysis on the HCP dataset, we train on 65 randomly selected subjects, monitor training on 15 validation subjects, and test on 20 subjects. In contrast with the DiSCo dataset, we estimated 3 group-wise tissue response functions, accounting for white matter, gray matter, and cerebrospinal fluid present in in-vivo human brains.

**Validation details.** In-vivo datasets lack microstructure ground truth. Some approaches [62] suggest using the CSD estimated fODFs from high-quality and high-resolution dMRI as a surrogate ground truth. However, we argue that this introduces a bias towards CSD and hinders the assessment of performance improvements in new methodologies. A more recent alternative proposed by [19, 20] involves using partial volume estimation (PVE) from fODFs as a validation metric. Nevertheless, the PVE scalar is rotation invariant, posing limitations on its ability to discern the impact of rotation-equivariant methods compared to their non-equivariant counterparts. Non-equivariant models may demonstrate high accuracy in invariant scalar estimation while failing to capture the underlying geometric structure adequately. Moreover, PVE validation relies heavily on the accuracy of third-party tissue segmentation on T1/T2 images, which may not consistently correlate with partial volume estimates derived from more information-rich but low-resolution dMRI data. While we acknowledge that a comprehensive solution for quantitative analysis of in-vivo fODF estimation is yet to be established, we advocate for a visual qualitative analysis of our proposed method. This approach aims to provide insights into the performance of our model without relying on potentially biased or limited validation metrics.

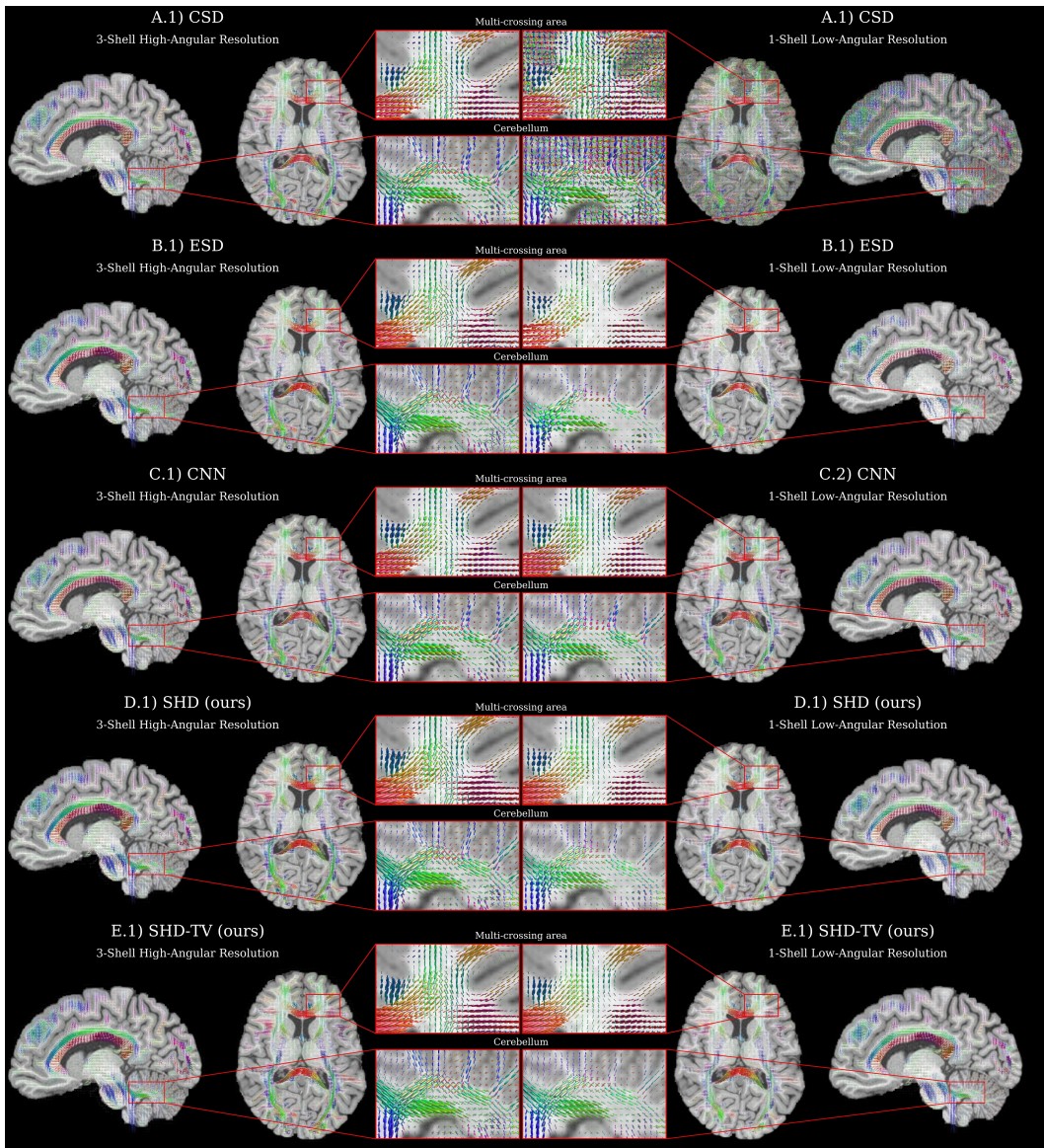

Figure 9: Qualitative illustration comparing the proposed equivariant dMRI deconvolution framework against all other baselines.

## A.5 Preliminary analysis on an abnormal brain

Here, we demonstrate the preliminary applicability of our method on abnormal brains. In Fig. 10, we estimate fODFs on a brain with a glioma using the recently released dMRI dataset from [75]. We find that the additional priors from our method help fODF estimation and fiber tracking substantially, filling in a hole in the fiber tracks recovered using the baseline CSD method. However, we note that this analysis is highly preliminary and that tractography on brains with lesions is a highly active area of research that requires substantial modifications to tractography algorithms [77].

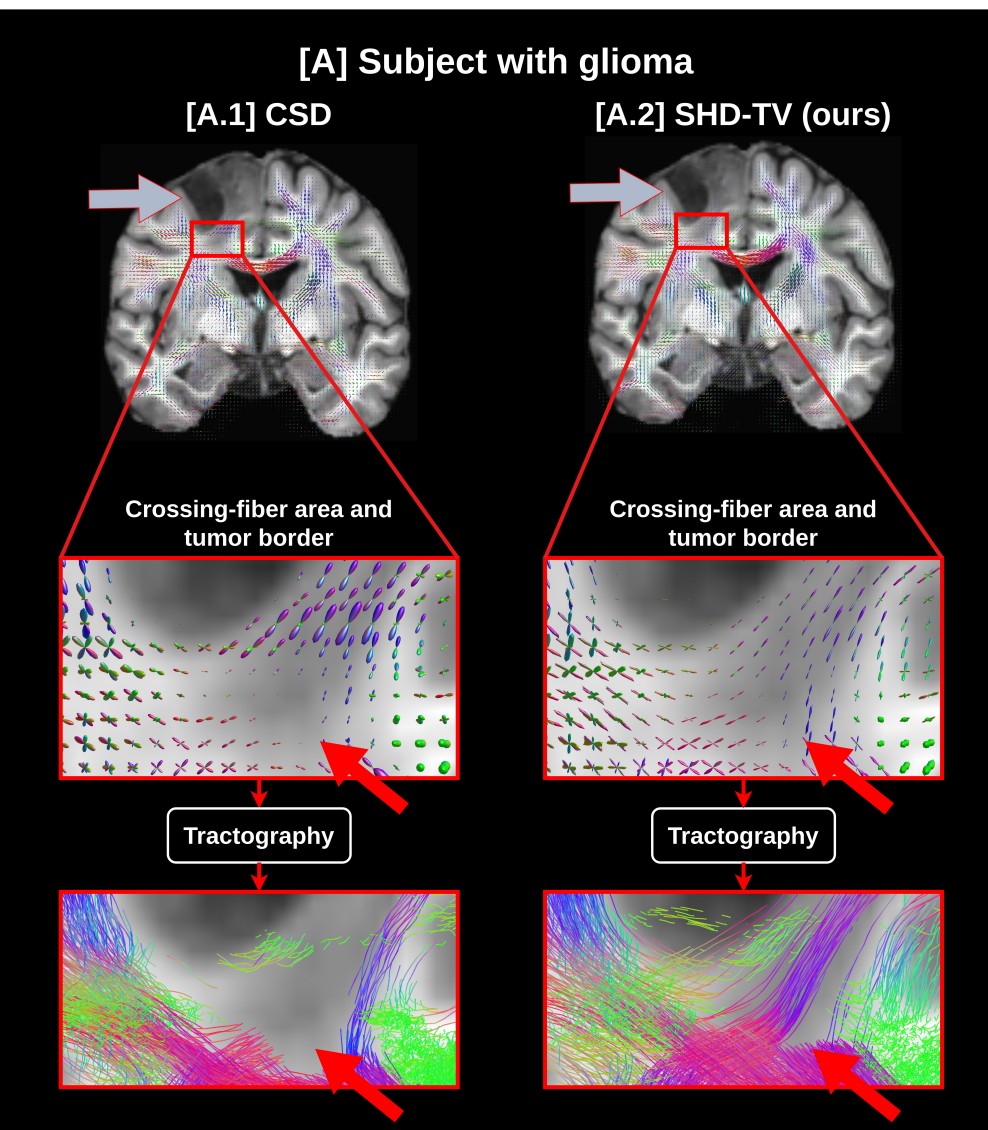

Figure 10: **fODF and tractography estimation in a glioma-affected brain** (gray arrow). In **[A]**, we compare the conventional CSD method (**[A.1]**) with our proposed SHD-TV model (**[A.2]**). Our approach retrieves more spatially coherent fODFs with better fiber angular separation in voxels containing crossing fibers. Notably, the CSD method does not detect fODFs in the crossing area indicated by the red arrow, leading to an inadequate representation of microstructures by the tractography algorithm. Additionally, our model does not reveal any abnormal fODFs near or within the tumorous tissue.

# B  Equivariance vs. Data Augmentation: $\mathbb{R}^3 \times \mathcal{S}^2$ MNIST

**Motivation.** We study the advantages of using spatio-spherical inductive biases for convolutional layers, as opposed to learning them from the data. We create a synthetic spatio-spherical MNIST dataset (inspired by the spherical MNIST experiments of [15]) to conduct controlled experiments.

**Dataset.** The $\mathbb{R}^3 \times \mathcal{S}^2$ MNIST data generation process is presented in Fig.11. Importantly, the dataset characterizes a spatio-spherical segmentation task, where voxel-wise information alone is insufficient for correct voxel classification. We construct two versions of this dataset to test generalization from either equivariance or data augmentation. The first version lacks any form of data augmentation, whereas the second version incorporates random grid and voxel-wise rotations. Because on-the-fly data augmentation of spatio-spherical signals is computationally expensive, we pre-generate a total of 1000 volumes per dataset, with a training-validation-test split of 716/142/142.

**Dataset generation.** We randomly position eight non-overlapping $4 \times 4$ squares on a 2D $16 \times 16$ slice. Subsequently, each square is assigned a random digit between 1 and 9, designating 0 as the background digit. This 2D slice is duplicated along the $z$-axis, yielding the final 3D segmentation volume. Notably, this volume comprises eight non-overlapping $4 \times 4 \times 16$ structures, termed *tubes*, aligned along the $z$-axis, each featuring the same classification digit across its voxels. Optionally, we apply a random grid rotation to the segmentation volume. In the second step, we randomly select an MNIST image corresponding to the classification digit assigned to each square. These digit images are projected onto a sphere using the methodology presented in [15]. We utilize HEALPix spherical sampling with a resolution of 4, equivalent to a spherical resolution of $V = 192$ vertices. Given the straightforward nature of voxelwise spherical digit classification, we reduce the amount of information at each voxel. Instead of projecting the entire MNIST image onto the sphere, we randomly crop it to one-quarter of its original size before projecting the cropped digit onto the sphere. As a result, any segmentation network has to rely on both the spatial neighborhood and the spherical information within a voxel. Optionally, we apply a random voxel rotation to the spherically projected digit.

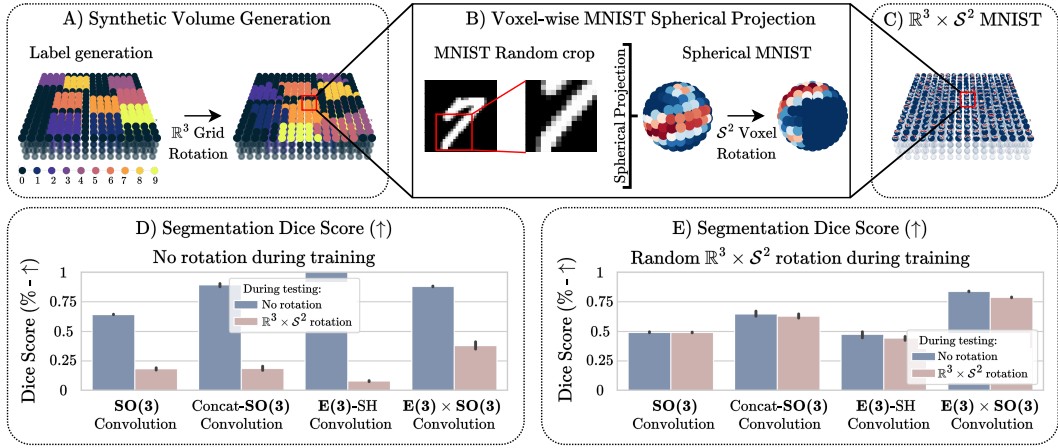

Figure 11: **Top row: Spatio-Spherical MNIST generation process. Bottom row: Results from testing generalization from equivariance vs. data augmentation on the synthetic $\mathbb{R}^3 \times \mathcal{S}^2$ MNIST classification task. [A-C]** Dataset generation process. **[D-E]** Segmentation dice score of models trained without (D) and with (E) data augmentation. Overall, our proposed $\mathbf{E}(3) \times \mathbf{SO}(3)$ convolution has a higher generalization power to unseen transformation than its non-equivariant counterparts, and the right inductive bias increases segmentation performance against data augmented-only models.

**Model details.** We compare four convolutional architectures, each exhibiting different equivariance properties while sharing the same U-Net structure presented in Fig.3. We compare the three previously presented spatio-spherical $\mathbf{E}(3) \times \mathbf{SO}(3)$, $\mathbf{E}(3)$-SH, and Concat-$\mathbf{SO}(3)$ U-Nets, and, to highlight the importance of incorporating spatial information, we compare them with a voxel-wise $\mathbf{SO}(3)$ U-Net. Inputs to the U-Nets consist of $16 \times 16 \times 16 \times 192$ $\mathbb{R}^3 \times \mathcal{S}^2$ MNIST volumes, and the outputs,

post-Softmax activation, are $16 \times 16 \times 16 \times 10$ volumes representing the classification probabilities of each voxel for the 10 digits.

**Training details.** Our training regimen uses a batch size of $8$ and trains for $50$ epochs with an initial learning rate of $3 \times 10^{-3}$, which is halved after the $25$, $35$, and $45$ epochs. The optimized objective function encompasses a combined Dice and Cross-Entropy loss with class-dependent weight to address imbalanced labels.

**Results.** We first assess how equivariance influences the generalization of models to unseen data transformations. The four models are first trained on the dataset without data augmentation and subsequently tested on the dataset incorporating out-of-distribution random rotations. We then train and test the models on the dataset augmented with random rotations. The results are presented in Fig.11.D) and E). To quantify segmentation performance, we compute the dice score across every volume in the testing set.

In Fig.11.D), we observe that all four models, regardless of the embedded equivariance, perform well when trained and tested on a transformation-free (i.e., in-distribution) dataset. Importantly, the spatially informed networks outperform the voxel-wise $\mathbf{SO(3)}$ network, underscoring the significance of incorporating spatial context into the neural network. Furthermore, we emphasize the importance of incorporating inductive bias into the neural network to enhance generalization to unseen data transformations during training. The $\mathbf{E(3)} \times \mathbf{SO(3)}$ model, trained on the rotation-free dataset, achieves a relatively better dice score of $0.38$ on the dataset with unseen rotations than both other models with only equivariance to $\mathbf{E(3)}$ or $\mathbf{SO(3)}$.

Fig.11.E) further illustrates the advantage of inductive bias compared to data augmentation during training. Data augmentation dramatically enhances the performance of the $\mathbf{E(3)} \times \mathbf{SO(3)}$ model on randomly rotated data, surpassing its non-equivariant counterparts. Meanwhile, data augmentation applied to $\mathbf{E(3)}$ or $\mathbf{SO(3)}$ models lacking spherical or spatial inductive bias improves segmentation performance but falls short of matching the performance of the $\mathbf{E(3)} \times \mathbf{SO(3)}$ equivariant model.

## C Additional experimental details

### C.1 Spatio-Hemispherical Convolution overview

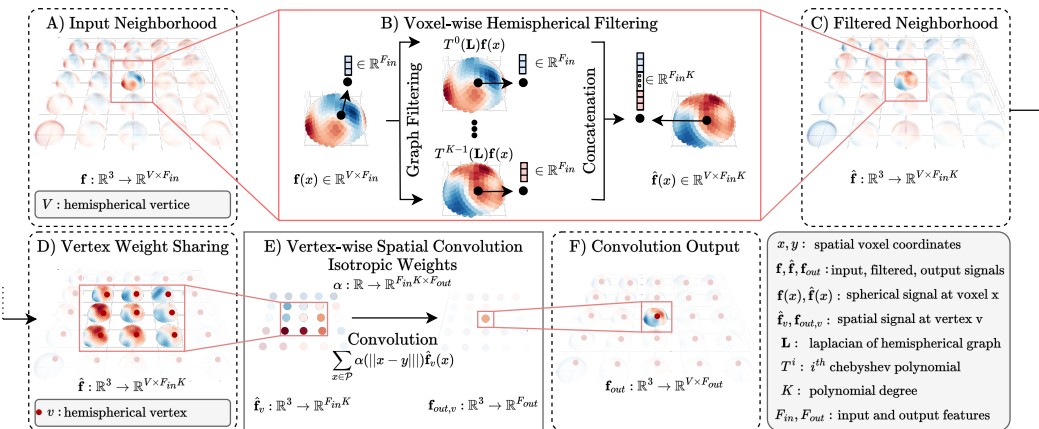

Figure 12: **Overview of the Hemispherical $E(3) \times SO(3)$ Convolution computation**. This figure is adapted from [20]. **[A-C]** The input of the convolution is a 3D grid of hemispherical graphs with $V$ vertices per voxel. The input is first processed by voxel-wise spherical filtering using the proposed hemispherical Laplacian and efficient implementation. **[D-F]** The 3D volume is then processed by a 3D isotropic convolution with weight-sharing across the hemispherical graph vertex.

### C.2 Details of the deconvolution framework

**fODF model.** The fODF model assumes the voxel-wise diffusion signal to be produced by a composition of $T$ tissue types, with each tissue further composed of an unknown number of cells/fibers, each one described by its local partial volume and orientation. At the voxel level, the fiber information is aggregated into one fiber orientation distribution function (fODF) per tissue, providing information on the local tissue composition and orientations. The model relates the dMRI signal and tissue microstructure by combining $T$ tissue-specific Response Functions (RFs) and $T$ fiber Orientation Distribution Functions (fODFs). The fODFs $F=\{F_t\}$ are tissue-dependent antipodally symmetric spatio-spherical functions. The response functions $RF=\{RF_t^b\}$ are tissue-dependent and shell-dependent spherical functions, assumed constant across the spatial domain, and to be antipodal and axial symmetric. The total $b$-shell dMRI signal is modeled as a sum over tissue-specific signals $S^b = \sum_{t=1}^T S_t^b$, where tissue-specific diffusion signals are modeled as a voxel-wise spherical convolution $S_t^b = \mathcal{C}(F_t, RF_t^b)$ between the fODF and the response function. The convolution is defined as $\mathcal{C}(F, RF)(x, q) = \int_{p \in \mathcal{S}^2} F(x, p) RF(q^T p) \mathrm{d}p$.

**Diffusion sampling.** A dMRI image $S$ is a function $S(x, q) : \mathbb{R}^3 \times \mathcal{S}^2 \to \mathbb{R}^B$. In practice, the dMRI signal is discretized on a set of shells $\mathcal{B} = \{b_i \in \mathbb{R}^+\}_{i \in [1,..,B]}$ where $B$ is the number of shells, such that for every $b \in \mathcal{B}$, the $b$-shell function $S^b$ is a spatio-spherical signal. Furthermore, the $b$-shell function is sampled on a shell-dependent set of spherical gradient direction $\mathcal{V}_b = \{q_i^b \in \mathcal{S}^2\}_{i \in [1,..,N_b]}$ with $N_b$ the number of $b$-shell gradient sampling. We note $\mathbf{S}_{\mathbf{b}, \mathcal{V}_b}(x) = [S(x, q_1^b, b), ..., S(x, q_{N_b}^b, b)] \in \mathbb{R}^{N_b}$ the sampled $b$-shell signal on $\mathcal{V}_b$. Moreover, we note $\mathcal{D} = \{\mathcal{B}, \{\mathcal{V}_b\}_{b \in \mathcal{B}}\}$ the full diffusion sampling set, and $\mathbf{S}_{\mathcal{D}}(x) = [\mathbf{S}_{\mathbf{b_1}, \mathcal{V}_{\mathbf{b_1}}}(x), ..., \mathbf{S}_{\mathbf{b_B}, \mathcal{V}_{\mathbf{b_B}}}(x)] \in \mathbb{R}^{N_\mathcal{D}}$ the sampled dMRI signal, with $N_\mathcal{D} = \sum_{b \in \mathcal{B}} N_b$ the total number of gradient sampling.

**Data normalization.** For every new dMRI scan $S^{ori.}$, we compute the scan-specific white matter B0 response function $RF_{wm}^{0,ori.}$ using MRtrix [68], and normalize the dMRI signal $S = S^{ori.}/RF_{wm}^{0,ori.}$. The input of our framework is the normalized shell-sampled diffusion MRI signal $\mathbf{S}_\mathcal{G}(x) = [\mathbf{S}_{\mathbf{b_1}, \mathcal{V}_{\mathbf{b_1}}}(x), ..., \mathbf{S}_{\mathbf{b_B}, \mathcal{V}_{\mathbf{b_B}}}(x)] \in \mathbb{R}^{N_\mathcal{D}}$ for $x \in \mathbb{R}^3$. Following [19], we first normalize the protocol-dependent spherical sampling by first interpolating it, for each diffusion shell, to a fixed hemisphere HEALPix sampling $\mathcal{V}^+$ as defined in section 3.2. We implement the interpolation using spherical harmonics. For this, we first compute the spherical harmonic coefficients $\hat{\mathbf{S}}_{\mathbf{b}}(x) \in \mathbb{R}^{N_L}$ for every

$b \in \mathcal{B}$ shell, where $N_L$ is the number of harmonic coefficient for a maximal harmonic bandwidth of $L$. Because the diffusion signal is antipodal-symmetric, the odd-order spherical harmonic coefficients are equal to $0$ and we only compute the even-order coefficients, making the total number of coefficient $N_L = (L/2 + 1)(L + 1)$. Importantly, getting the coefficients $\hat{\mathbf{S}}_{\mathbf{b}}(x)$ from $\mathbf{S}_{\mathbf{b},\mathcal{V}_b}(x)$ requires at least $|\mathcal{V}_b| \geq (L/2 + 1)(L + 1)$ samples. We limit the maximum interpolation bandwidth to $L = 8$, which requires at least $45$ samples per shell. In case the shell sampling $\mathcal{V}_b$ does not have enough samples, we use the theoretically maximum spherical harmonic degree estimated from the available gradients. We then interpolate the harmonic coefficients on the HEALPix grid $\mathcal{V}^+$, $\mathbf{S}_{\mathbf{b},\mathcal{V}^+}(x) = \hat{\mathbf{S}}_{\mathbf{b}}(x)\mathbf{Y}_{\mathcal{V}^+}^L$, where $\mathbf{Y}_{\mathcal{V}^+}^L$ relates the even $L$ bandwidth harmonic coefficients to the HEALPix spherical sampling. We get the $B$ feature input spatio-spherical signal $\mathbf{S}_{\mathcal{V}^+}(x) \in \mathbb{R}^{V \times B}$.

**Group-average response function.** We estimate a population-based response function $\hat{\mathbf{R}}\mathbf{F}$ following [55]. For $N_s$ normalized dMRI scans in our training dataset, we first compute $T$ tissue response functions $\hat{\mathbf{R}}\mathbf{F}_{t,i}$ per dMRI image, where $i$ is the image index, using the Dhollander algorithm, implemented in MRtrix [68] [18]. We then compute the per-tissue average response function $\hat{\mathbf{R}}\mathbf{F}_{\mathbf{t}} = 1/N_s \sum_{i=1}^{N_s} \hat{\mathbf{R}}\mathbf{F}_{t,i}$.

**Signal reconstruction.** From the fODFs $\mathbf{F}_{\mathcal{V}^+}$, we reconstruct the diffusion signal $\mathbf{S}_{\mathcal{D}}$ on the reconstruction shell-sampling $\mathcal{D}$ using a spherical convolution with response functions $\mathbf{RF}$. For every voxel $x \in \mathbf{R}^3$, we compute the spherical harmonic coefficients $\hat{\mathbf{F}}(x) = \{f_{l,t}^m(x)\} \in \mathbb{R}^{N_L \times T}$ from the estimated $\mathbf{F}_{\mathcal{V}^+}(x)$. The fODFs being antipodal-symmetric, we only compute the even-degree coefficients. The maximum spherical harmonic degree $L$ depends on the resolution of the HEALPix sampling $\mathcal{V}^+$. We use a sampling resolution with $384$ vertices per hemisphere and a fODF maximum spherical harmonic degree of $L = 18$. Moreover, the response functions are also represented by their spherical harmonic coefficients $\hat{\mathbf{R}}\mathbf{F} = \{r_{l,t}^{m,b}\} \in \mathbb{R}^{M_L \times T \times B}$. The RFs are antipodal and $z$-axis symmetric, setting to $0$ every non-even degree and non-zero order coefficients. We then compute the reconstruction dMRI spherical harmonic coefficients $\hat{\mathbf{S}}(x) = \{s_l^{m,b}(x)\} \in \mathbb{R}^{N_L \times B}$, $s_l^{m,b}(x) = \sum_t \sqrt{2\pi/(2l+1)} f_{l,t}^m(x) r_{l,t}^{0,b}$. Finally, we can interpolate the reconstructed signal $\hat{\mathbf{S}}$ on any shell-sampling $\mathcal{D}$.

### C.3 Details of compared models

We give an overview of details of the compared methods in Fig.13 and separate the different baseline methodologies into two groups.

**Conventional Models.** Conventional methods for fODF estimation generally solve the inverse problem iteratively. Constrained Spherical Deconvolution [66, 68] (CSD) and RUMBA [11] optimize voxel-wise dMRI reconstruction losses subject to a non-negativity constraint on the fODF. RUMBA-TV further enhances fODF smoothness by incorporating spatial total variation regularization. Importantly, conventional methods do not need prior training and are limited by the input data quality.

**Deep-Learning Models.** We evaluate our proposed framework against existing deep-learning methodologies for fODF estimation. While diverse network architectures, such as MLP, 2D/3D CNN, or spherical CNN, have been proposed, we adopt a standardized network architecture to focus our comparison on the inductive bias and training framework introduced by each method's convolution layer. We employ an MLP-based network that operates voxel-wise [52] and a CNN-based network operating on spatial patches [46]. Adding spherical inductive biases to the MLP network, we employ a voxel-wise $\mathcal{S}^2$-based network ESD [61, 19]. The $\mathbf{SE}(3)$-equivariance PONITA method adds spatial information and inductive bias, equivariant to joint transformation on the spatio-spherical domain. Finally, the RT-ESD model is a $\mathbf{E}(3) \times \mathbf{SO}(3)$-equivariance spatio-spherical convolution network, equivariant to independent transformation on the spatial and spherical domain. We extend RT-ESD to an efficient implementation SHD and spatially regularized SHD-TV.

### C.4 Details of architecture and training

**Network Details.** A high-level network architecture is illustrated in Fig. 3. For $\mathbf{E}(3) \times \mathbf{SO}(3)$-equivariant models, we use spatio-spherical pooling and unpooling operations. The $\mathbf{SO}(3)$-equivariant models employ spherical pooling/unpooling, the CNN models use only spatial pool-

ing/unpooling, and no pooling is used by the MLP model. The first layer maps the input features to $F_s = 32$ features for equivariant models and $F_s = 128$ features for non-equivariant models.

The input and output feature number of the first and last layers depend on the convolution inductive bias, the number of estimated tissue components $T$, the number of input diffusion shells $B$, and the diffusion gradients angular resolution. The MLP and CNN models have at each voxel a feature vector of size $F_i = B \times 45$ (high angular resolution) or $F_i = B \times 28$ (low angular resolution) consisting of the 45 or 28 spherical harmonic coefficients of degree 8 or 6, respectively. At each voxel, they output a feature vector of size $F_o = T \times 45$, representing the 8-degree spherical harmonic coefficients of the $T$ tissue fODF. The $S^2$ U-Net, ESD, and RT-ESD models use an input spherical graph of 768 vertices (HEALPix grid resolution of 8) with $F_i = B$ spherical feature maps. The Concat-ESD model uses the same graph resolution but the number of feature maps increases to $F_i = B \times 3^3$ due to neighborhood concatenation. All four models output $F_o = T$ spherical maps (with the same resolution), corresponding to the $T$ tissue fODFs.

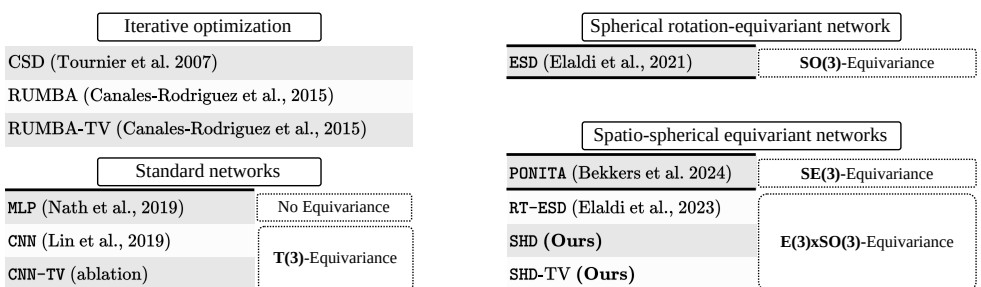

Figure 13: **Model description.** We compare conventional and learnable methods. Learnable methods have increasing embedded geometric prior through equivariance.

**Training implementation.** All models are trained for 50 epochs with a batch size of 16 patches using the Adam optimizer [41] with an initial learning rate set at $1.7 \times 10^{-2}$. The learning rate is decayed by a factor of ten after the 30, 40, and 45 epochs. For unsupervised models, we tuned the regularization weights using the ESD model on the DiSCo validation volume, and further tune $\lambda_{tv}$ for SHD: $\lambda_{nn} = 10^{-1}$, $\lambda_{sparse} = 5 \times 10^{-5}$, and $\lambda_{tv} = 5 \times 10^{-1}$. The spatially-informed models operate on $3 \times 3 \times 3$ spatial patches, whereas the spherical-only models process $1 \times 1 \times 1$ spatial patches.

**fODF ground truth generation.** Quantitative validation of the model requires access to ground truth fODF. The two benchmark datasets either do not directly provide this information or provide reference fODFs estimated with the CSD model on noise-free high-angular resolution dMRI volumes. However, both datasets provide ground-truth white matter streamlines. To avoid bias towards the CSD model and extract ground-truth fODF for the Tractometer dataset, we propose an alternative unbiased fODF estimation approach leveraging the ground-truth tractograph. We approximate the voxel-wise ground-truth fODFs by aggregating every streamline passing through a voxel and computing its spherical fiber density function. A white matter streamline can be represented as a set of vectors $\{v_j^i\}$ where $v \in \mathbf{S}^2$ is the local streamline direction, $j$ is the streamline index and $i$ is the vector index. For every voxel $p$, we find every local direction $\{v_{j_k}^{i_k}\}$ going through the voxel. We then apply a spherical kernel density estimation on the set of unit vectors $\{v_{j_k}^{i_k}\}$ using a uniform spherical kernel with an angular size of $15°$. We then extract the ground-truth peak directions from the fiber density function using the MRtrix peak detection algorithm [68] with a maximum number of crossing fibers per voxel set to 10.

