# OpenReview forum: "Equivariant spatio-hemispherical networks for diffusion MRI deconvolution"
_NeurIPS.cc/2024/Conference — NeurIPS 2024 poster_

### Official Review · Reviewer_rCgS · 2024-07-07

**Soundness:** 2
**Presentation:** 2
**Contribution:** 2
**Rating:** 3
**Confidence:** 4

**Summary:**

The paper presents a convolutional neural network for spherical deconvolution of DWI data to estimate fiber orientation distribution. The main contributions over previous approaches include the introduction of Spatio-Hemispherical Equivariant Convolution, Dense Matrix Multiplication, and the use of Pre-computed Chebyshev Polynomials. These innovations improve overall efficiency. Evaluation was conducted on simulated data, assessing efficiency in terms of memory and runtime, and accuracy, demonstrating reduced false positive rates and angular error.

**Strengths:**

1. Enhanced efficiency for DWI spherical deconvolution using deep neural networks.
2. Improved accuracy compared to current methods, demonstrated with simulated data designed for such experiments.
3. Provided code for reproducibility.

**Weaknesses:**

1. Limited Novelty: The primary innovations over previous approaches focus on the network's efficiency (Section 3.2). Consequently, the main factors contributing to the reported improvement in accuracy remain unclear.
2. Clarity of Presentation: The paper is highly detailed, making it difficult to follow and understand the main contributions that actually lead to improvements in performance (Section 3.2) and accuracy.
3. Limited Demonstration of Impact: The practical impact beyond accuracy on simulated Tractometer data is not clearly demonstrated. It remains uncertain whether the proposed approach offers any significant clinical or scientific applications.
4. Mixed Results: Figure 6D is particularly disappointing, as the results produced by the proposed method noticeably differ from the reference.

**Questions:**

1. Please clearly describe the main novel contributions of the paper and how they contributed to the results.
2. Please thoroughly discuss Figure 6D, where the results of the proposed approach appear to deviate substantially from the reference.

**Limitations:**

The authors discuss the limitations of the paper. However, the lack of demonstration of clinical applications and the generalization to multiple clinical/scientific DWI acquisition settings should be discussed more thoroughly.

---

> ### Author Rebuttal · Authors · 2024-08-07
>
> Thank you for the valuable feedback and for highlighting our improved efficiency and accuracy in comparison to current work!
>
> We believe that there may have been some miscommunications on our part and hope to clarify them below:
>
> # Unclear reasons for improvement
> > _"Limited novelty as presentation focused on efficiency. The main factors contributing to the reported improvement in accuracy remain unclear."_
>
> To clarify, the accuracy gains primarily come from using physically motivated priors to solve highly ill-posed fODF reconstruction problems.
>
> To recap, previous work either
> - Does **voxel-by-voxel** iterative optimization (CSD) or unsupervised equivariant network training (ESD), accounting for only the spherical nature of diffusion signals.
> - More recent work (RT-ESD) does unsupervised equivariant network training accounting for both spatial and spherical symmetries. However, it is **too computationally intense** to run practically, which is half of our motivation.
>
> Our accuracy gains come from two perspectives,
> 1. We now correctly model the **antipodal symmetry** of diffusion signals, broadly seen in clinical in vivo dMRI, and directly incorporate it into our network kernels. This allows the network to focus on the reconstruction task instead of also trying to learn the antipodal symmetry of the data. This convolution is detailed in Section 3.2/”Spatio-hemispherical equivariant convolution”.
> 2. Further, we incorporate a physically-motived regularizer encouraging low **spatial total variation** into the unsupervised network training. To our knowledge, this regularizer has not been used in previous deep dMRI deconvolution networks. Its inclusion significantly improves fiber localization over previous methods, particularly when working with clinically used low-angular resolution images.
>
> Both contributions, alongside the low-level network kernel analyses, are entirely novel and contribute to the observed efficiency and accuracy gains.
>
> We would be happy to provide any additional information on these contributions will directly emphasize these perspectives in the revision.
>
>
> > _"The paper is highly detailed, making it difficult to follow and understand the main contributions that actually lead to improvements in performance and accuracy."_
>
> Thank you for raising this concern. We have outlined the main contributions that led to the reported gains in the answer above.
>
> Regarding the contribution presentation, we will incorporate your feedback (and that of Reviewer `zoNj`) to revise the presentation to be easier to follow, thank you for raising this concern.
>
> For reference, the information regarding the causes of performance increases are currently presented in the following sections:
> - The theoretical and technical explanation of the improved efficiency is provided in Section 3.2, with a quantitative analysis of each contribution detailed in Section 4.1 and illustrated in Fig. 4.
> - Section 3.3 covers the contributions to improved fODF estimation accuracy. A detailed quantitative analysis of our method is presented in Section 4.2 and depicted in Figs. 6 and 7, emphasizing the importance of total variation regularization during training.
>
> We will streamline this in the revision.
>
> # Experiments
> > _"The practical impact beyond accuracy on simulated Tractometer data is not clearly demonstrated. It remains uncertain whether the proposed approach offers any significant clinical or scientific applications."_
>
> We respectfully disagree. Our experiments include quantitative analyses on two benchmark datasets widely used by the dMRI community and also include qualitative analyses on in vivo HCP data, widely used in a variety of dMRI papers, including at NeurIPS \[[1](https://arxiv.org/pdf/2011.01355),[2](https://arxiv.org/pdf/2306.00854)\].
>
> For context, fiber orientations in human diffusion MRI at clinical resolution have no ground truth. Acquiring this ground truth would require post-mortem dissection-based analyses and we are not aware of any publicly available dMRI dataset of this kind. As a result, the dMRI analysis community focuses on highly realistic simulated benchmark datasets to measure algorithmic progress. This is consistent with the challenge datasets of DiSCo and Tractometer used in our paper, where we achieve state-of-the-art fODF reconstruction results at more practical speeds.
>
> Further, our method is entirely generic and can be plugged into any existing dMRI analysis pipeline for analyzing the connectivity of the human brain. Lastly, the most accurate of previous work (RT-ESD) was unsuitable for clinical and scientific applications due to its computational load. Instead, our work is highly scalable due to its efficiency gains and can be practically deployed and matches or exceeds its accuracy.
>
> We would welcome any further discussion on this matter and will clarify the text to emphasize these points.
>
> > _"Fig6D is disappointing, results produced by the method noticeably differ from the reference"_
>
> To clarify, Fig. 6D is a qualitative result of a single deconvolved voxel at clinical resolution. The actual **dataset-wide results are presented in Fig. 6B**, wherein we achieve much fewer false positives with lower angular error when looking at the entire dataset.
>
> For context, this is a highly undersampled dataset that is challenging for all baselines and we cannot expect reconstructions from low-angular resolution to match the performance on high-angular resolution. However, the baseline CSD produces an entirely different fiber orientation and ESD and CNN produce spurious fibers. Only SHD-TV has the correct number of major fibers with a lower angular error to the ground truth.
>
> Again, these are highly undersampled inputs so no method will achieve perfect reconstruction due to the ill-posedness, but dataset-wide, we measure a 30% decrease in angular error, and a 27% decrease in missing estimated fibers (false negative rate), when using SHD-TV in comparison to CSD, for example.

---

> > ### Comment · Reviewer_rCgS · 2024-08-11
> >
> > Thank you for your response.
> >
> > The comments and discussion provided by the authors indeed clarify their contribution. Yet, the contribution is very specific to dMRI. It is probably a better match to a dMRI / Neuroimaging-related conference than to the broader audience in neurips. I therefore keep my score.

---

> ### Author Response · Authors · 2024-08-12
>
> Thank you for engaging in the discussion phase!
>
> We are happy to see that the reviewer has no remaining technical and experimental concerns w.r.t. their original review. Their new concern pertains purely to scope.
>
> > _“Yet, the contribution is very specific to dMRI. It is probably a better match to a dMRI / Neuroimaging-related conference than to the broader audience in neurips.”_
>
> We respectfully disagree and believe that there may be a misunderstanding as,
>
> ``
> ### **1. NeurIPS has already published several papers on dMRI**
>
> dMRI has been explored extensively by both machine learning and neuroscience researchers due to its rich geometric structure and use in measuring neural connectivity.
>
> As a result, NeurIPS has already featured several papers on dMRI analysis with machine learning, demonstrating its relevance to the NeurIPS audience. For example,
> - [NeurIPS’23](https://proceedings.neurips.cc/paper_files/paper/2023/file/294de0fa7149adcb88aa3119c239c63e-Paper-Conference.pdf)
> - [NeurIPS’20](https://proceedings.neurips.cc/paper/2020/file/bc047286b224b7bfa73d4cb02de1238d-Paper.pdf)
> - [NeurIPS’19](https://papers.nips.cc/paper_files/paper/2019/file/0bfce127947574733b19da0f30739fcd-Paper.pdf)
> - [NeurIPS’17](https://proceedings.neurips.cc/paper/2017/hash/ccbd8ca962b80445df1f7f38c57759f0-Abstract.html)
> - [NeurIPS’14](https://proceedings.neurips.cc/paper_files/paper/2014/file/215a71a12769b056c3c32e7299f1c5ed-Paper.pdf)
>
> Additionally, similar conferences and journals in machine learning and computer vision have all featured machine learning work on dMRI data, further supporting its broad relevance to the machine learning community. For example,
> - [ICLR’23](https://openreview.net/forum?id=0vqjc50HfcC)
> - [CVPR’24](https://openaccess.thecvf.com/content/CVPR2024/html/Fadnavis_Patch2Self2_Self-supervised_Denoising_on_Coresets_via_Matrix_Sketching_CVPR_2024_paper.html)
> - [PAMI’22](https://www.computer.org/csdl/journal/tp/2022/02/09247263/1oslcBeZ3l6)
>
>
> ``
> ### **2. Our work is a general geometric deep learning contribution for spatio-spherical data**
>
> NeurIPS/ICML/ICLR and similar venues are strongly interested in geometric deep learning and deep learning on manifolds. However, such manifold structure only arises in specialized applications that may seem niche at first but later become of wide interest to the machine learning community (e.g. geometric deep learning for [molecular docking](https://openreview.net/forum?id=kKF8_K-mBbS)).
>
> While we focus on dMRI in our paper, our work is generically beneficial to the analysis of spatio-spherical signals as it finds several avenues for efficiency gains and builds a framework for sparse non-negative spatio-spherical deconvolution. We foresee several potential benefits in contexts where spatio-spherical data arises: [robotics](https://www.roboticsproceedings.org/rss14/p23.pdf), [neural rendering](https://openaccess.thecvf.com/content/CVPR2022/papers/Fridovich-Keil_Plenoxels_Radiance_Fields_Without_Neural_Networks_CVPR_2022_paper.pdf), gaussian splatting, [molecular dynamics](https://openreview.net/forum?id=dPHLbUqGbr), etc.
>
> ``
> ### **3. NeurIPS invites interdisciplinary work in its call for papers**
>
> NeurIPS explicitly encourages interdisciplinary submissions in its [Call for Papers](https://neurips.cc/Conferences/2024/CallForPapers).
>
> Our work lies at the intersection of the core “_Machine learning for sciences (life sciences)_” and “_Neuroscience and cognitive sciences_” areas mentioned in the call as it directly contributes:
> - New geometric equivariant deep learning methods for a core life sciences imaging modality (dMRI).
> - A novel self-supervised non-negative deconvolution formulation on a spatial graph of spherical signals.
> - Enhanced neuronal fiber recovery, which is crucial for the core neuroscience task of understanding brain connectivity.
>
>
> ``
> ### **4. Neuroscience is a core topic at NeurIPS and dMRI is the main tool for understanding neural connectivity**
>
> Neuroscience has been central to NeurIPS from its inception and understanding the [structural connectivity](https://www.sciencedirect.com/science/article/pii/S1053811913005351) of the brain _in vivo_ relies entirely on dMRI, among having other applications such as [surgical planning](https://www.nature.com/articles/s41593-024-01570-1).
>
> Therefore, in addition to the geometric deep learning community, we foresee our work being of interest to NeurIPS’ neuroscience community as well as our work contributes new deep learning methods that significantly advance the analysis of such data.  In turn, these deconvolution advancements lead to more accurate neural pathway estimation that can potentially improve downstream neuroscientific and biomedical analyses.
>
> Thanks again for your engagement!

---

### Official Review · Reviewer_F5CU · 2024-07-12

**Soundness:** 4
**Presentation:** 4
**Contribution:** 3
**Rating:** 7
**Confidence:** 4

**Summary:**

The authors extend previous [work done](https://proceedings.mlr.press/v227/elaldi24a.html) in the diffusion MRI (dMRI) fibre orientation distribution function (fODF) domain with an efficient $\mathbf{E}(\mathbf{3}) \times \mathbf{SO}(\mathbf{3})$ equivariant network. The proposed model directly leverages the antipodal symmetry of dMRI data to reduce computation time by 65%, as compared to previous work. The authors demonstrate the efficacy of this approach in a number of experiments; including an analysis of fODF estimation in simulated data and real world data, as well as in a downstream tractography task. They find that their method often performs best, whilst maintaining relatively high compute and memory efficiency.

**Strengths:**

The writing and format of this work is excellent. The attention to detail, as provided within the main text and the appendix, rivals that of full length journal articles within this domain.

**Weaknesses:**

This work represents a continuation of a previous method developed within [Elaldi et al](https://proceedings.mlr.press/v227/elaldi24a.html). Whilst the authors here present a significant increase in computational efficiency, this study is an iterative improvement on previous approaches, rather than a leap forward. I caveat that by acknowledging the importance of iterative improvements within scientific research.

Overall, the writing is of an excellent standard. However, I have a small number of suggestions/mistakes enumerated below.

- Line 98 you state that "trainable models have the advantage of decreasing the reliability of the method...". Here, _reliability_ evokes “reliable, as in you can count on it” rather than “rely on, as in this is a prerequisite”. I would maybe switch to “need for” or similar.
- Line 151 I think you're missing a word at the end of the sentence "sparse matrix multiplication significant computational..."
- Line 162 "Fig. 3 overviews", I would use "presents an overview" rather than using overview as a verb.
- Line 269 I think "unsupervisedly" sounds a little clunky, would swap for "in an unsupervised manner" or similar.
- Line 270 you state "...to extract fODFs and then use the estimated fODF to investigate the effect of improved local fODF estimation on...". I think this could be reworked to use the words fODF and estimat(ed/ion) a little less, perhaps by swapping "the estimated fODF" for "them", or swap "investigate the effect of improved local fODF estimation" for "investigate their effect".

**Questions:**

Given that your experiments either involve simulated data or healthy patient data, when tasked with reconstructing fODFs for subjects with significant brain pathologies, would you reasonably expect to see a drop in performance? Or do you suspect that the regularisation enforced via the equivariant properties and loss functions would be enough such that the difference in performance would be minimal?More generally, how would you expect this method to perform when tasked with prediction on out-of-distribution data, as compared to the iterative per-subject CSD method?

**Limitations:**

The authors have adequately addressed the limitations of their work

---

> ### Author Rebuttal · Authors · 2024-08-07
>
> Thank you for the encouraging feedback! Broadly, we will incorporate all of the detailed suggestions and address the remaining high level questions/comments below:
>
> > _”Whilst the authors here present a significant increase in computational efficiency, this study is an iterative improvement on previous approaches, rather than a leap forward. I caveat that by acknowledging the importance of iterative improvements within scientific research.”
>
> We agree that we are building extensively on previous work, in particular the spatio-spherically equivariant RT-ESD method. However, RT-ESD is too computationally intensive for scalable use in either research or the clinic.
>
> As a result, we develop a principled framework that leverages a physically motivated assumption about dMRI to build new equivariant layers that retain the equivariance, but are much faster to compute and store in memory. Further, we conducted an extensive analysis of these network kernels and identified key avenues for improvement (eg, pre-computing the Chebyshev polynomials) in order to make deep learning on very high dimensional dMRI data tractable.
>
> We agree that this is not necessarily a paradigm shift methodologically, however, our contributions have significant potential for real-world application which was not possible with previous work.
>
> > _"I have a small number of suggestions/mistakes enumerated below."_
>
> We greatly appreciate the detailed enumeration of typos and awkward phrasings! We will make all suggested changes.
>
> > _"[...] When reconstructing fODFs for subjects with significant brain pathologies, would you reasonably expect to see a drop in performance? [...] how would you expect this method to perform on out-of-distribution data [...]?"_
>
> Thank you for raising this question. We do agree that our model can be further validated under more diverse clinical conditions. As there is no _in-vivo_ microstructural ground truth for the human brain, especially for brains with pathologies, validating novel methods is challenging and ill-posed.
>
> However, as testing data distributions shift, we believe our additional proposed priors benefit ill-posed reconstruction. Our priors of spatial smoothness, spatio-spherical equivariance, sparse fibers, and antipodally symmetric signals are all valid for brains with pathologies as well. We therefore do not expect degradation on brains with lesions.
>
> As a preliminary exploration, in Fig. 1 of the rebuttal PDF, we now perform an initial fODF analysis on a brain with a tumor using the recently released dataset from \[[1](https://www.nature.com/articles/s41597-024-03013-9)\]. We find that the additional priors from our method help fODF estimation and fiber tracking substantially yielding smooth fibers, whereas the baseline CSD method has a hole in its estimated fiber tracks.
>
> However, we emphasize that this analysis is preliminary and that tractography on brains with lesions is a highly active area of research that requires substantial modifications to tractography algorithms \[[2](https://www.sciencedirect.com/science/article/pii/S1053811921009241)\], which are outside the scope of our fODF estimation work. We will mention this as a limitation and area for future work and add these new results to the appendix.

---

> > ### Comment · Reviewer_F5CU · 2024-08-11
> >
> > Thank you for your response.
> >
> > The comments and discussion provided by the authors is sufficient, and I therefore keep my positive score.

---

### Official Review · Reviewer_zoNj · 2024-07-26

**Soundness:** 3
**Presentation:** 3
**Contribution:** 3
**Rating:** 6
**Confidence:** 3

**Summary:**

This work introduces a novel framework for fODF estimation through equivariant spatio-hemispherical networks that achieve dMRI deconvolution. Experiments on simulated dMRI datasets with known ground truth, as well as on real in vivo dMRI data are conducted, showing promising results while improving over previous methods.

**Strengths:**

1. The paper improves upon previous methods in both processing time, and quantitative results.
2. The evaluation is sound and the experiments nicely show results on synthetic datasets with known ground truth.

**Weaknesses:**

The main weakness of this manuscript is in the way the contributions section is written (at the end of the Introduction section, lines 57--71).
The authors would potentially increase readability of their paper by making this paragraph as clear and as sound as possible.
It would also help readers quickly identify if they wish to continue reading this paper and if it is of interest to their own research.

My suggestions are:
1) Introduce this paragraph by restating what the main aims of this paper are (similar to what you wrote on lines 20--23).
2) Clearly introduce the technical contributions as they are backed by the experiments / results section with a short description of what was achieved.
3) Some of the contributions (specifically, the in vivo qualitative results) are not present in the main manuscript, but are part of the appendix. The exception is Figure 2 which does not have enough description in the main text (lines 254--256), and appears in the middle of a paragraph discussing the synthetic data results. I believe that the experiments section should be clearly reflected in the main aims and contributions of the paper, and the appendix should be used for optional / additional results which do not take away from the main contributions. I understand that there is a limit of 9 content pages to the paper, and I am happy to discuss this further.

**Questions:**

Please find below some questions and general suggestions:

1. Figure 1 introduces the readers to an example of how dMRI data looks, and it is an important prelude towards understanding the problem statement of your paper. For this reason, I suggest the authors include further explanations in this figure, in either visual form or in the captions, such as:
    1. How does gradient 25 differ from gradient 288 (maybe try to explain / show that these are different gradient directions and/or strengths instead of the 1/25/288 indices which have no specific meaning in this context)?
    2. I think it is also important to show a zoomed-in version of the T1w image, to not confuse the reader that the spatio-spherical signal is also present in the structural data.

2. Please be consistent with referencing your figures in the manuscript: you sometimes write Figure x, and sometimes write Fig. x

3. I suggest you introduce the name of your proposed spatial-hemispherical deconvolution (SHD) framework in the contributions section (lines 57--71) as on the next page Figure 2 shows examples of your model.

4. In Figure 6B, could you discuss whether the low-resolution input to high-resolution output experiments could produce unrealistic reconstructions in the presence of noise / in a real dMRI data setting, as high-angular resolution is needed for higher contrast in the angular domain? I am wondering if for crossing fibers, for example, as shown in Figure 6D, none of the methods can accurately reconstruct the ground truth then maybe we cannot trust these reconstructions for low-angular resolution data?

5. Can you also please label the x-axes in Figures 6A and 6B to make it clear that the values are in degrees?

6. In Figure 6C can you explain the “narrowness” of your result as compared to the ground truth or CSD?

7. Can you provide a short description (in section 4.2.1) of how the peak angular error and false positive rates are calculated? I understand that these are described in A.3, but to improve readability I suggest that they are introduced a bit sooner with the details left for the appendix, or at least to make it more clear that the details are in A.4. Moreover, it would be interesting to understand the slight increase in FPR in both Figures 6A and 6B when comparing SHD(-TV) with RT-ESD.

**Limitations:**

The authors have attempted to address some potential limitations, but would be nice to see a lengthier discussion on how their proposed method would perform on other in vivo clinical datasets, under different noise levels, patient motion, etc.

---

> ### Author Rebuttal · Authors · 2024-08-07
>
> Thank you for the highly detailed and valuable feedback! We will incorporate the suggestions and address the other concerns below:
>
> # Weaknesses
>
> > _"The main weakness of this manuscript is in the way the contributions section is written at the end of the Introduction. [...]"_
>
> Thank you for the detailed suggestions on editing this paragraph! We will incorporate them into the revised paper.
>
> > _"Some of the contributions (specifically, the in vivo qualitative results) are not present in the main manuscript, but are part of the appendix."_
>
> As the reviewer mentions, we combined the qualitative in-vivo fODF estimation and quantitative synthetic analysis in Section 4.2.1, as both address the same question. We clarify that this section details both results and is not solely focused on quantitative outcomes.
>
> However, we agree that our complete qualitative and quantitative analysis was not entirely contained in the main paper due to space limitations. In our revision, we will expand the caption of Fig. 2 and bring in more qualitative in vivo results into the main paper from the appendix. To do so, we will abbreviate the front matter of the introduction and move some lower level experimental details to the appendix.
>
> # Questions
>
> > _"Could you discuss whether the low-resolution input to high-resolution output experiments could produce unrealistic reconstructions in the presence of noise / in a real low-resolution dMRI data setting [...]?"_
>
> Thank you for raising this important discussion point. We agree with the reviewer that hallucination is a concern in all undersampled reconstruction methods, including previously proposed fODF estimation methods.
>
> As demonstrated in our experiments, in low angular settings, the current widely-used Constrained Spherical Deconvolution (CSD) method provides inaccurate fiber estimates with high angular error. Our motivation is precisely to mitigate these unrealistic reconstructions by considering the spatial correlation of the diffusion signal and leveraging the network's equivariance property.
>
> In our DiSCo experiments with clinically relevant noise, as shown in Fig.6.B, using both the spatially-informed network and spatio-spherical equivariance leads to a significant decrease in hallucinated false positive fibers and angular error. **Importantly, the FPR achieved with our method on low-angular resolution input is competitive with the results on high-angular resolution input**.
>
> However, we entirely agree that undersampled reconstruction results must be approached with caution, as the lower angular resolution increases angular error and the likelihood of missing estimated fibers, which can be qualitatively observed in Fig.2.B. This caution should be considered within the broader context of the tissue microstructure estimation field. For example, Diffusion Tensor Imaging, which uses only a few spherical samples to estimate the underlying fiber configuration, can sometimes create spurious local reconstructions but remains a primary tool in clinical applications and research. Our method, with its additional priors and self-supervised learning strategy, represents a significant improvement over methods currently used in clinical settings.
>
> We will clarify the results to include this discussion.
>
> > _"[...] lengthier discussion on how the proposed method would perform on other in vivo clinical datasets, under different noise levels, motion, etc."_
>
> We agree that this is an important discussion. In the attached rebuttal PDF, we perform an additional experiment using dMRI images of brains with pathologies \[[1](https://www.nature.com/articles/s41597-024-03013-9)\]. As also suggested by reviewer `JrFU`, we agree that our method would benefit from further specific validation under varying image qualities and the presence of artifacts.
>
> Briefly, we find that additional priors of spatio-spherical equivariance and spatial smoothness improve fiber estimation on in vivo anomalous clinical data, with the caveat that such analyses need to be performed on much wider scale in a clinical followup for certainty. Further, we believe that as image quality degrades, that is precisely where additional priors help for ill-posed reconstruction. Due to space limitations in the rebuttal, please see our discussion with Reviewers `JrFU` and `F5CU` for further details. These limitations and avenues for further work will be added to the revision.
>
> > _"[...] slight increase in FPR in both Fig 6A and 6B when comparing SHD(-TV) with RT-ESD."_
>
> The reviewer is correct that there is a slight increase in the FPR.
> Regarding the SHD versus RT-ESD comparison, we speculate that as FPR and Angular Error have a tradeoff, they might require slightly differing regularization weights. SHD also has the prior of antipodal symmetry whereas RT-ESD does not, and the bias-variance tradeoff might be causing this slight increase.
> For the proposed SHD-TV model, the increase in FPR can be attributed to the smoothing effect of the total variation regularization, which occasionally extends a fiber into neighboring voxels where it might not be appropriate. However, this increase in FPR is offset by the benefits of much lower angular error and higher spatial coherence, which enhances robustness against noise and improves fiber localization.
>
> > _"In Fig6C can you explain the “narrowness” of your result as compared to the ground truth or CSD?"_
>
> We use a prior of sparse fibers using a sparse regularizer during training, this is consistent with previous work such as ESD and RT-ESD and the wider fODF literature. This leads to increased narrowness in comparison to the CSD method, which does not use any sparsity regularization.
>
> # Writing improvement suggestions
>
> Thanks again for all of the detailed writing and presentation suggestions. We will incorporate all of them into the revision.

---

> > ### Comment · Reviewer_zoNj · 2024-08-09
> >
> > Thank you for your response.
> >
> > The clarifications added by the authors to all questions raised have addressed most of my concerns. I therefore keep my positive score.

---

### Official Review · Reviewer_JrFU · 2024-07-28

**Soundness:** 3
**Presentation:** 3
**Contribution:** 3
**Rating:** 6
**Confidence:** 4

**Summary:**

This paper introduces a novel method for analyzing diffusion MRI data, leveraging convolutional network layers equivariant to the E(3)×SO(3) group, which respects the physical symmetries of dMRI data. The proposed spatio-hemispherical graph convolutions reduce computational complexity while maintaining high deconvolution accuracy.

**Strengths:**

This paper presents a novel method for dMRI deconvolution by introducing equivariant convolutional network layers that account for the physical symmetries in dMRI data. The use of spatio-hemispherical graph convolutions, leveraging the antipodal symmetry of neuronal fibres, reduces computational complexity while maintaining accuracy. The proposed method addresses important challenges in dMRI analysis, focusing on the need for accurate deconvolution at clinically feasible resolutions.

The theorical foundation and empirical validation is sufficient, and the methodology is well-presented with clear explanations. The results are consistently validated, showcasing the method's efficiency and accuracy improvements. Additionally, The clarity of the paper is good with well-organized structure.

**Weaknesses:**

1. The reliance on specific assumptions may limit the scope of the model. It would be better to improve the flexibility of the model so that it could be applied to diverse scenarios.

2. Diverse clinical conditions can be considered in future studies, involving varying levels of image quality and pathological changes. Conditions such as specific noise or artifacts are not fully explored.

**Questions:**

1. Can the authors clarify the limitations of the antipodal symmetry assumption? Are there specific scenarios where this assumption might not hold, and how might this impact the model's performance?

2. Can the authors discuss the generalizability of their method across different clinical conditions and patient populations? How adaptable is the model to varying clinical data qualities?

**Limitations:**

Yes.

---

> ### Author Rebuttal · Authors · 2024-08-07
>
> Thank you for the positive evaluation and for highlighting the importance and quality of our methodology, presentation, and experiments.
>
> # Antipodal symmetry assumption
>
> > _"It would be better to improve the flexibility of the model so that it could be applied to diverse scenarios."_
>
> We agree that the antipodal spherical symmetry assumption limits our approach to antipodally symmetric spatio-spherical data. While previous methods \[[1](https://arxiv.org/pdf/2310.02970),[2](https://arxiv.org/pdf/2304.06103),[3](https://arxiv.org/pdf/2102.06942)\], have developed spatio-spherical equivariant networks without this assumption, they are far too computationally intensive to scalably process high-dimensional data such as diffusion MRI. We believe that this is partly why the diffusion MRI community primarily still uses conventional iterative methods.
>
> We instead build on previous works by incorporating more knowledge of the underlying task directly into the network layers to significantly increase computational efficiency. However, we agree with you that future work should aim to relax these assumptions while maintaining our efficiency gains and this is mentioned in our future work.
>
> > _"Can the authors clarify the limitations of the antipodal symmetry assumption?"_
>
> Thank you for initiating this important discussion! The antipodal symmetry assumption is widespread in dMRI analysis due to the symmetric nature of the diffusion process. Consequently, only a symmetric fODF can be estimated from a single voxel-wise diffusion signal, and **mainstream deconvolution methods widely use this assumption**. For instance, the most widely-used fODF estimation method, Constrained Spherical Deconvolution \[[4](https://www.sciencedirect.com/science/article/abs/pii/S1053811907001243)\], computes only the even-order spherical harmonics of the fODF, thus making the antipodal symmetry assumption.
>
> The one limitation is that at a _microscopic_ level, fODFs can have antipodal asymmetry. However, this is only visible in ex vivo dissection studies and we focus on in vivo clinical dMRI where antipodal symmetry is a valid assumption. This assumption is also reflected in common dMRI acquisition strategies in large-scale studies that only sample hemispherical signals at every voxel \[[5](https://www.ncbi.nlm.nih.gov/pmc/articles/PMC8317510/),[6](https://onlinelibrary.wiley.com/doi/10.1002/mrm.21646),[7](https://www.ncbi.nlm.nih.gov/pmc/articles/PMC7065087/)\]. We will mention this in the Discussion of the revised paper.
>
> # More diverse clinical conditions
>
> > _"Diverse clinical conditions can be considered in future studies, involving varying levels of image quality and pathological changes [...]."_
>
> Thank you for suggesting this possible future direction! We do agree that our model can be further validated under more diverse clinical conditions. However, as there is no _in-vivo_ microstructural ground truth for the human brain, validating novel methods is challenging and ill-posed.
>
> In our paper, we rely on community-developed standardized benchmarks and datasets. As mentioned by the reviewer, our paper presents comprehensive experiments to quantitatively validate the performance improvements of our method by performing:
> - fODF estimation analyses with diverse image qualities, quantitatively on the DiSCo and Tractometer datasets, as well as qualitatively on the HCP dataset.
> - Downstream tractography task analyses on the Tractometer dataset.
> - Speed and memory analyses.
>
> Regarding the specific scenarios mentioned by the reviewer,
>
> ## Brains with lesions
> In Fig. 1 of the rebuttal PDF, we perform an initial fODF analysis on brains with anomalies using the recently released dataset from \[[8](https://www.nature.com/articles/s41597-024-03013-9)\]. We find that the additional priors from our method help fODF estimation and fiber tracking substantially, filling in a hole in the fiber tracks recovered using the baseline CSD method.
>
> However, we note that this analysis is preliminary and that tractography on brains with lesions is a highly active area of research that requires substantial modifications to tractography algorithms \[[9](https://www.sciencedirect.com/science/article/pii/S1053811921009241)\], which are outside the scope of our fODF estimation work. We will mention this as a limitation and area for future work and add these new results to the appendix.
>
> ## Varying image qualities
>
> Regarding image qualities, all of the datasets in our paper have substantially different characteristics and SNRs. Further, we believe our model generalizes well across **different clinical conditions and patient populations** because:
> - It is trained to reconstruct the input diffusion signal directly, rather than recognizing specific tissue structures that may or may not be present at clinical deployment.
> - The self-supervised training framework allows the model to be fine-tuned for any new image and imaging setup.
>
> We agree that testing performance under varying amounts of patient motion and imaging artifacts would be highly interesting. However, we are not aware of any publicly available in vivo datasets that provide such data. Our limitations and future work will now further emphasize that future work should investigate performance under various amounts of head motion or other clinical artifacts.

---

> > ### Comment · Reviewer_JrFU · 2024-08-11
> >
> > Thank you for the response. The rebuttal has addressed most of my concerns. I will keep my scores.

---

### Author Rebuttal · Authors · 2024-08-07

We thank the reviewers for their time and their encouraging feedback. Their comments and suggestions have made the revision a stronger paper.

We were happy to find that they found the submission to be theoretically sufficient \[`JrFU`\], well presented and organized \[`JrFU, F5CU`\], extensively validated with sound evaluation \[`JrFU`, `zoNj`\], yielding better efficiency and accuracy \[`JrFU`,`F5CU`,`rCgS`\], with a high attention to detail \[`F5CU`\].

Common questions and concerns revolve around the clarity of our contributions paragraph in the Introduction \[`zoNj`, `rCgS`\], our antipodal symmetry assumption \[`JrFU`\], the limited analysis in the main text of the in-vivo experiments \[`zoNj`\], and a lack of clinical applications \[`rCgS`\]. The reviewers would also like to see a longer discussion of our method in more diverse clinical settings, such as pathological changes, noise, and motion  \[`JrFU`, `zoNj`, `F5CU`\].

## Changes and clarifications

To improve our submission and address the reviewers' concerns, we make the following major revisions and clarifications, briefly recapped below:
-  \[`JrFU`\] We motivate the physical reasons behind the assumption of antipodal symmetry of diffusion MRI signals and describe the computational benefits below.
-  \[`zoNj`,`rCgS`\] We will use the suggestions to rewrite our contribution paragraph in the revision so as to make the paper accessible and clearly understandable.
-  \[`zoNj`,`rCgS`\] We clarify our experiments on the highly undersampled qualitative example visualized in Fig 6D.
-  \[`zoNj`,`rCgS`\] We provide additional descriptions of our quantitative and qualitative results, regarding the narrowness of the predictions, slight increases in false positive fibers on a specific experiment, and the challenges of prediction on low-angular resolution undersampled images.
-  \[`JrFU`,`zoNj`,`F5CU`\] We now provide a qualitative analysis of the outcome of our method on dMRI of a brain with pathologies and discuss how our model adapts to new clinical and imaging setups.

All details and remaining comments are addressed in the individual responses below. Additional results are attached as a PDF.

We appreciate the reviewers' comments on our contribution and thank them again for their time and expert feedback. We would be happy to address any additional concerns and incorporate any further suggestions.

---

### Decision · Program_Chairs · 2024-09-25

**Decision:**

Accept (poster)

**Comment:**

Two reviewers recommend Weak Accept, another reviewer recommends Accept and a fourth reviewer recommends Reject. The main argument of reviewer rCgS to recommend Reject is that the subject of the paper is better suited for a "dMRI / Neuroimaging-related conference than to the broader audience in neurips". The authors rightly provide a battery of good arguments as to why a paper such as this has a place at NeurIPS. I fully agree with the authors. Furthermore, the paper goes beyond the simple application of a method to the dMRI domain and proposes a new equivariant deep-learning architecture which is particularly designed for the peculiarities of dMRI data. I therefore recommend Accept.